# Tumor-infiltrating nerves functionally alter brain circuits and modulate behavior in a mouse model of head-and-neck cancer

Jeffrey Barr[1†], Austin Walz[1†], Anthony C Restaino[1,2], Moran Amit[3], Sarah M Barclay[1], Elisabeth G Vichaya[4], William C Spanos[1,2], Robert Dantzer[3], Sebastien Talbot[5], Paola D Vermeer[1,2]*

[1]Sanford Research, Cancer Biology and Immunotherapies Group, Sioux Falls, Sioux Falls, United States; [2]University of South Dakota, Sanford School of Medicine, Vermillion, United States; [3]University of Texas, MD Anderson Cancer Center, Houston, United States; [4]Baylor University, Department of Psychology and Neuroscience, Waco, United States; [5]Queen's University, Department of Biomedical and Molecular Sciences, Kingston, Canada

*For correspondence:
paola.vermeer@sanfordhealth.org

[†]These authors contributed equally to this work

Competing interest: The authors declare that no competing interests exist.

**Abstract** Cancer patients often experience changes in mental health, prompting an exploration into whether nerves infiltrating tumors contribute to these alterations by impacting brain functions. Using a mouse model for head and neck cancer and neuronal tracing, we show that tumor-infiltrating nerves connect to distinct brain areas. The activation of this neuronal circuitry altered behaviors (decreased nest-building, increased latency to eat a cookie, and reduced wheel running). Tumor-infiltrating nociceptor neurons exhibited heightened calcium activity and brain regions receiving these neural projections showed elevated Fos as well as increased calcium responses compared to non-tumor-bearing counterparts. The genetic elimination of nociceptor neurons decreased brain Fos expression and mitigated the behavioral alterations induced by the presence of the tumor. While analgesic treatment restored nesting and cookie test behaviors, it did not fully restore voluntary wheel running indicating that pain is not the exclusive driver of such behavioral shifts. Unraveling the interaction between the tumor, infiltrating nerves, and the brain is pivotal to developing targeted interventions to alleviate the mental health burdens associated with cancer.

## eLife assessment

This **important** research describes the sensory innervation of oral tumors, with potential implications for understanding cancer-induced alterations in motivation and anhedonia in a mouse model. These findings are **solid** and are supported by anatomical and transcriptional changes in the tumor that suggest sensory innervation, neural tracing, and neural activity measurements. While nerve innervation of the tumor and associated increase in brain activity is well-supported, future studies could enhance specificity by employing more targeted genetic and pharmacological tools to manipulate these circuits selectively.

## Introduction

The prevalence of mental health disorders (e.g. depression, anxiety, suicide) in cancer patients is significantly greater than in the general population (*Carreira et al., 2018*; *Pitman et al., 2018*; *Walker et al., 2013*). For those patients with no prior psychiatric history, a cancer diagnosis increases the risk of mental health decline (*Zhu et al., 2017*). The presence of cancer introduces many stressors

**eLife digest** A lot of cancer survivors experience a decline in mental health, persisting often decades after successful treatment. Many factors contribute to this reduced mental well-being, including the physical, emotional and financial stresses they experience.

Scientists think that the increased prevalence of mental health disorders among cancer patients and survivors may also be linked to the cancer itself. Previous research has shown that most tumors, in particular in melanomas, cervical and ovarian cancers, and head and neck cancers, contain sensory nerves that sense thermal, mechanical and chemical changes and so alert an organism about a potential danger, such as extreme temperature, pressure, changes in pH or inflammation.

To investigate whether these nerves contribute to the worsened mental health of cancer patients, Barr, Walz et al. studied male mice with tumors growing in their mouths, mimicking the disease of patients with head and neck cancers. The mice with tumors exhibited several altered behaviors linked to their well-being, suggesting that they had reduced overall health compared to mice without tumors. For example, they were less inclined to build nests, accept treats or run on a wheel.

Next, Barr, Walz et al. injected a fluorescent dye into the tumors to label the nerves inside the cancerous growths. Fluorescence microscopy and imaging studies revealed that, days later, the dye had traveled to multiple regions of the brain, indicating that the nerves in the tumors had connected to a preexisting nerve circuit that included these brain regions.

Further experiments revealed that the nerve cells in these brain regions were more active in mice with tumors and had different functional properties compared to mice without tumors. Removing the connecting nerves either genetically or with a drug improved all the behaviors of the mice with tumors. Treating the mice with painkillers also improved some but not all of their behaviors, indicating that pain is not the exclusive driver of such behavioral shifts. These two experiments suggest that the nerves from the tumors relay information about pain to the brain and contribute to reduced well-being of the mice.

Further studies will test whether these tumor-brain connections also contribute to behavioral changes in mice with other types of cancer. The data suggest that disrupting the neural connections between a tumor and the brain may improve the mental health of patients with cancer, but more research is needed to establish this link.

(physical, financial, relational) into the lives of patients, thus a negative impact on mental health may not be surprising. However, these changes persist even in long-term cancer survivors. For instance, decade-long cancer survivors maintain an increased incidence of depression (approximately 12%, depending on the cancer type) as compared to the population at large (3–5%; *Steel et al., 2014*; *Götze et al., 2020*; *Kuba et al., 2019*). While the intensity and prevalence of psychological symptoms in cancer patients fluctuate before, during, and after treatment for a given type of cancer, and between cancer types, it remains higher than in the general population (*Wang et al., 2020*; *Naser et al., 2021*; *Tsaras et al., 2018*). The association of cancer with impaired mental health is directly mediated by the disease, its treatment or both; these findings suggest that the development of a tumor alters brain functions.

We have demonstrated the presence of TRPV1-expressing nociceptor neurons in head and neck squamous cell carcinomas (HNSCC; *Madeo et al., 2018*), melanoma (*Balood et al., 2022*), cervical (*Lucido et al., 2019*), and ovarian cancers (*Barr et al., 2021*). Nerve recruitment to the tumor bed is an active process that involves the release of soluble factors, including neurotrophins (*Renz et al., 2018*; *Hayakawa et al., 2017*; *Wang et al., 2014*) and neuropeptides (*Kasprzak and Adamek, 2020*). Tumor-released small extracellular vesicles (sEVs) also recruit loco-regional nerves to the tumor bed (*Madeo et al., 2018*; *Silverman et al., 2021*; *Amit et al., 2020*). While these and other studies establish that solid peripheral tumors engage with the peripheral nervous system (*Madeo et al., 2018*; *Barr et al., 2021*; *Cole et al., 2015*; *Magnon et al., 2013*; *Sadighparvar et al., 2021*; *Chen and Ayala, 2018*), it raises the possibility of a direct neuronal connection from the tumor to the brain.

A recent study used pseudorabies virus and mapped a connection from tumor-infiltrating nerves in an orthotopic model of murine lung cancer to areas in the brain (*Chen et al., 2022*). We expand these findings and demonstrate that HNSCC-associated nerves are transcriptionally and functionally altered

and project to discrete regions in the brain. The brain neurons connected to the tumor manifest increased activity, which is associated with behavioral alterations in tumor-bearing animals. Consistent with this, newly diagnosed HNSCC patients suffer high rates of depression and anxiety and lower quality of life (*Henry et al., 2022*). As most cancer patients face cancer-related pain (*Virgen et al., 2022*; *Hjermstad et al., 2009*; *van den Beuken-van Everdingen et al., 2007*), we tested whether treating pain could restore normal behavior. While pain treatment restored normal function at the tumor site, such as nesting behavior, it only partially restored normal voluntary running wheel behavior. Our findings suggest that, in addition to pain, tumor-infiltrating nerves communicate signals to the brain that lead to cancer-associated changes in behavior.

## Results

### Tumor innervation begins early in disease

MOC2-7 cells are HNSCC cancer cells derived from a CXCR3 null mouse on a C57BL/6 background (*Judd et al., 2012*). Their implantation in male mice results in dense innervation of tumors with nociceptor nerves (*Restaino et al., 2023*). To define the timing of tumor innervation, C57BL/6 wildtype male mice were orthotopically implanted (oral cavity) with MOC2-7 cells, and tumors were collected on days 4-, 10-, and 20- post-implantation. Western blot analysis of whole tumor lysate indicated expression of Tau, a neuronal marker (*Georgieff et al., 1993*), as early as day four post-tumor implantation, which increased over time (*Figure 1A*, *Figure 1—source data 1*, *Figure 1—source data 2*, *Figure 1—figure supplement 1*). A similar increase in the neuronal marker doublecortin (DCX) was noted (*Figure 1B*, *Figure 1—source data 3*, *Figure 1—source data 4*, *Figure 1—figure supplement 2*). While DCX is well-known for its expression in immature neurons, it is also expressed in adult peripheral neurons, including dorsal root ganglia (*Dellarole and Grilli, 2008*), and in non-neuronal tissues (*Bernreuther et al., 2006*). The DCX signal did not originate from the tumors, as MOC2-7 cell lysate was negative for this protein (*Figure 1B*, *Figure 1—source data 3*, *Figure 1—source data 4*). Consistent with this, immunohistochemical staining of these tumors with the neuronal marker β-III tubulin (β3T), demonstrated the increasing presence of nerves beginning on day 4 post-tumor implantation (*Figure 1C–E*).

### Tumor-infiltrating nerves map to the ipsilateral trigeminal ganglion and into the CNS

To map the origin of tumor-infiltrating nerves and define the circuits they converge upon, mice with palpable oral MOC2-7 tumors (approximately day 15 post-tumor implantation) were intra-tumorally injected with wheat germ agglutinin, WGA, a neural tracer conjugated to a fluorophore (n=10 mice). WGA is a lectin molecule that specifically binds to sialic acid residues present ubiquitously on neuronal membranes. It has been used extensively to map neuronal circuits centrally and in the periphery (*Borges and Sidman, 1982*; *van der Want et al., 1997*) and is a known transganglionic and transynaptic neuronal tracer (*McNicholas and Michael, 2017*; *Dumas et al., 1979*; *Sillitoe, 2016*) making it ideal for mapping neural circuits (*Tabuchi et al., 2000*; *Levy et al., 2017*; *Itaya, 1987*; *Itaya et al., 1986*; *Carson and Mesulam, 1982*). Following tracer injection, tumor growth was permitted for an additional 3–7 days to allow time for tracer labeling to occur. Animals were then euthanized, and tumors, trigeminal (TGM) ganglia, and brains were harvested and analyzed by microscopy. Microscopic examination of tumors revealed nerves with robust WGA signals (*Figure 1F*) as well as the V3 branch of the ipsilateral, but not the contralateral, TGM ganglion (*Figure 1G and H*). Consistent with the restricted labeling of tumor-infiltrating nerves, the tracer did not diffuse outside the tumor mass (*Figure 1I*).

Examination of brains from MOC2-7 tumor-bearing animals revealed tracer⁺ neurons in specific regions including spinal nucleus of the trigeminal (SpVc), parabrachial nucleus (PBN), and central amygdala (CeA; *Figure 1J*). Sections of brain from these regions show the presence of tracer-positive neurons (*Figure 1K*). Tracer injections of equivalent volume and concentration into the oral cavities of control non-tumor-bearing animals did not label the TGM ganglia nor areas in the brain (data not shown). A larger volume (10 µl) of tracer injected into non-tumor bearing mice resulted in tracer labeling of ipsilateral TGM neurons and brain (*Figure 1L*). These control studies indicate that the nerve density and distribution present in the tumor bed are higher than in control mice, with the

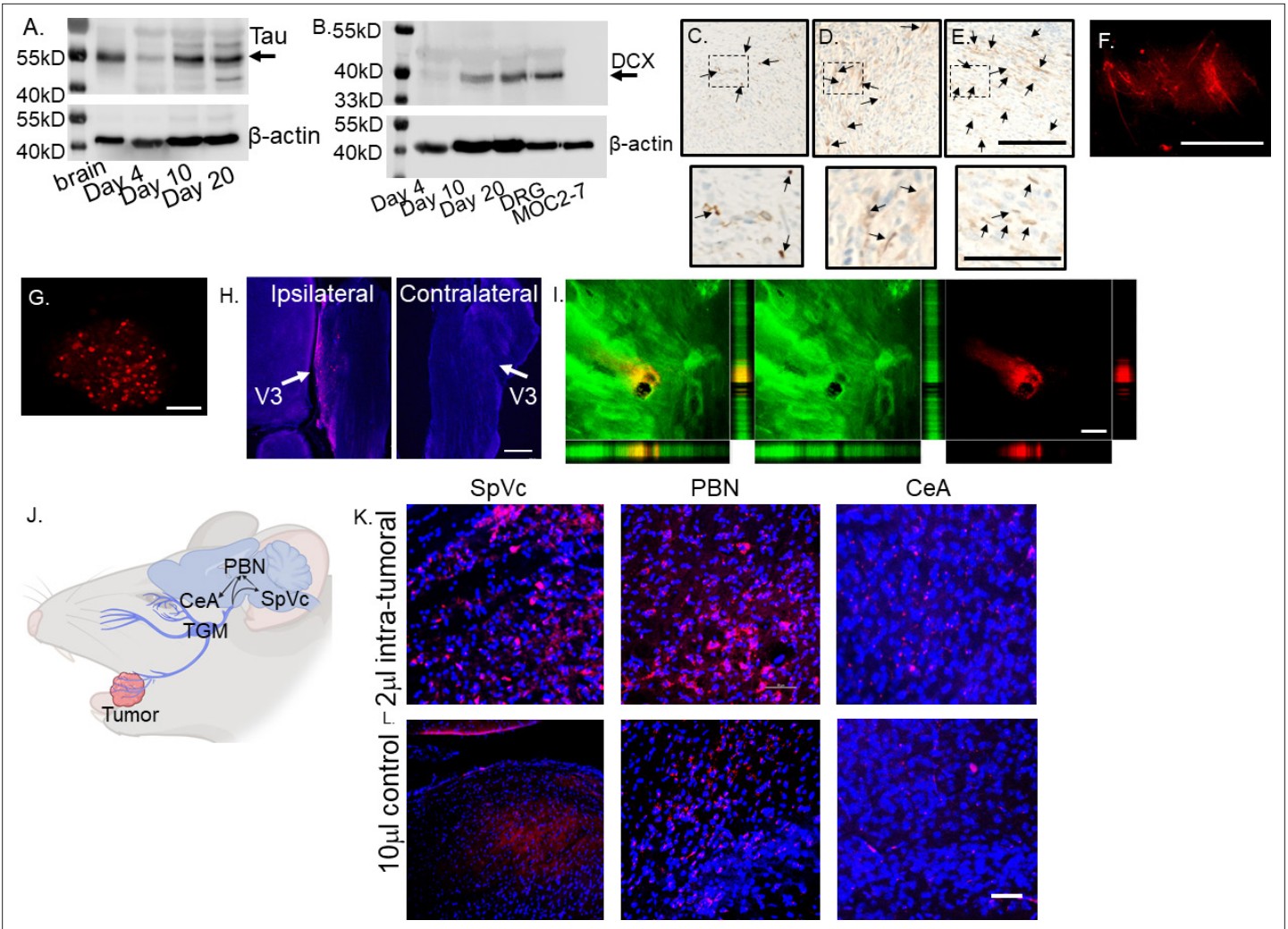

**Figure 1.** Tumor-infiltrating nerves form a circuit connecting to the brain. (**A**) Western blot of MOC2-7 whole tumor lysates for Tau (**A**) and Doublecortin, DCX (**B**). Tumors were harvested at different time points (as indicated) post-tumor implantation (n=3 mice/time point). Mouse brain, positive control; MOC2-7, whole cell lysate control; β-actin, loading control. Immunohistochemical staining for β-III tubulin (β3T, brown, arrows) of tumors harvested on days 4 (**C**), 10 (**D**) and 20 (**E**) post tumor implantation. Scale bar, 100 μm. Dotted boxes are shown at higher magnification below each panel; scale bar, 50 μm. Arrows highlight positive nerves staining. Representative *en face* confocal image of WGA positive (red) neurites within tumor (**F**) and the ipsilateral trigeminal ganglion (**G**) from a mouse orthotopically implanted with a MOC2-7 tumor and intra-tumorally injected with the tracer. Scale bars, 200 μm. (**H**) Low-magnification section of ipsilateral and contralateral TGM ganglia from a MOC2-7 tumor-bearing mouse that was intra-tumorally injected with WGA. The V3 branch of the TGM is marked with an arrow. WGA (red) positive neurons found only in the ipsilateral TGM. Scale bar, 500 μm. (**I**) Compressed Z-stack of confocal images demonstrating restricted WGA (red) within the tumor bed (cytokeratin, green). Vertical and horizontal cross-sections highlight that WGA (red) localizes strictly within the tumor (green) and does not leak out. Yellow shows the merged file. Scale bar, 500 μm. (**J**) Diagram showing the location of the trigeminal (TGM) ganglion, extension of its neurites into the tumor bed and the existing circuit which includes the spinal nucleus of the TGM (SpVc), the parabrachial nucleus (PBN) and the central amygdala (CeA) which get labeled following WGA injection into tumor. (**K**) Representative confocal images of the SpVc, PBN and CeA following injection of 2 μl of WGA into an oral MOC2-7 tumor (n=10 mice). Representative confocal images of SpVc, PBN and CeA following injection of (**L**) 10 μl (n=3 mice) of WGA into the oral cavity of a non-tumor bearing control animal. Scale bar, 50 μm. WGA (red looks pink due to DAPI), DAPI nuclear counterstain (blue).

The online version of this article includes the following source data and figure supplement(s) for figure 1:

**Source data 1.** Uncropped and labeled gels for *Figure 1a*.

**Source data 2.** Uncropped and labeled gel for *Figure 1a*.

**Source data 3.** Raw unedited gel for *Figure 1a*.

**Source data 4.** Uncropped and labeled gel for *Figure 1a*.

**Source data 5.** Uncropped and labeled gels for *Figure 1b*.

*Figure 1 continued on next page*

consequence that a small volume (2 µl instead of 10 µl) of tracer is sufficient to result in nerve labeling. The mapped circuit encompasses pre-existing connections to brain areas that regulate pain and affect (*Yeh et al., 2018*; *Hu, 2016*; *Zhu et al., 2022*; *Russo and Nestler, 2013*). These data indicate that the nerve infiltration into the tumor extends this circuit.

## Tumor-infiltrating neurons have altered transcript levels

The observation of nerves infiltrating the tumor mass prompted us to ask whether they undergo alterations by their mere presence within this 'foreign' environment. Consistent with this, we previously found that melanoma-infiltrating neurons have a unique transcriptome (*Balood et al., 2022*). To test whether this is also the case in HNSCC, we orthotopically implanted MOC2-7 cells into the oral cavity of male wildtype mice. After 14 days, TGM neurons were harvested and analyzed by qPCR using a commercial array for neural transmission and membrane trafficking genes. Compared to control TGM (non-tumoral) neurons, ipsilateral TGM neurons from tumor-bearing animals harbored increased expression (>fourfold) in genes involved in neuronal signaling/receptors (*Gabrg1*, *Gabra4*, *Grin2c*, *Grm3*) and synaptic transmission (*Gria2*; *Figure 2A*; *Supplementary file 1*; *Michetti et al., 2019*). The gene demonstrating the highest increase in expression, *Fus*, was of particular interest; it increases in expression within DRG neurons following nerve injury and contributes to injury-induced pain (*Han et al., 2023*; *Du et al., 2022*). Of note, we purposefully used whole trigeminal ganglia rather than FACS-sorted tracer-positive dissociated neurons to avoid artificially imposing injury and altering the transcript levels of these cells (*Liu et al., 2022b*; *Malin et al., 2007*). Thus, significantly elevated expression of *Fus* by ipsilateral TGM neurons from tumor-bearing animals suggests the presence of neuronal injury induced by the malignancy. This is consistent with our previous findings (*Inyang et al., 2024*) and those of others (*Horan et al., 2022*) showing that tumor-infiltrating nerves harbor higher expression of nerve-injury transcripts and neuronal sensitization.

## Tumor-infiltrating neurons are functionally changed

Given this transcriptomic alteration, we tested whether tumor-infiltrating neurons exhibit functional alterations. Thus, MOC2-7 tumor-bearing mice were intra-tumorally injected with the fluorophore-conjugated WGA tracer. Five days later, tracer+ TGM ganglia were cultured, and responsiveness to noxious stimuli was analyzed by calcium microscopy. Neuronal $Ca^{+2}$ responses to capsaicin (300 nM), which binds and activates TRPV1 channels, were measured by Fluo-4AM fluorescence microscopy. We found that tracer+ tumor-infiltrating ipsilateral TGM neurons show increased responsiveness to capsaicin compared to tracer negative contralateral neurons (amplitude, *Figure 2B–D*; the area under the curve, *Figure 2E*). Similar to contralateral TGM neurons, non-tumor-bearing TGM neurons elicited a normal response to capsaicin (*Figure 2F*).

This heightened sensitivity could reflect increased TRPV1 expression and/or its phosphorylation. We tested whether this was the case using western blotting and discovered an overexpression of the sigma 1 receptor (σ1R) and increased TRPV1 phosphorylation in the TGM ganglia from tumor-bearing animals (*Figure 2G*, *Figure 2—source data 1*, *Figure 2—source data 2*, *Figure 2—source data 3*, *Figure 2H and I*). Of note, the σ1R is an endoplasmic reticulum chaperone protein that directly interacts with TRPV1 and regulates its membrane expression (*Ortíz-Rentería et al., 2018*), while protein kinase C phosphorylation of serine 502 and 800 of TRPV1 reduces the receptor's activation threshold (*Bhave et al., 2003*; *Numazaki et al., 2003*; *Numazaki et al., 2002*; *Bhave et al., 2002*). Taken together, our data indicate that HNSCC-infiltrating nerves have a unique transcriptome and a heightened sensitivity to noxious stimuli characterized by an increased TRPV1 expression and phosphorylation consistent with TRPV1 sensitization secondary to oral cancer (*Scheff et al., 2022*).

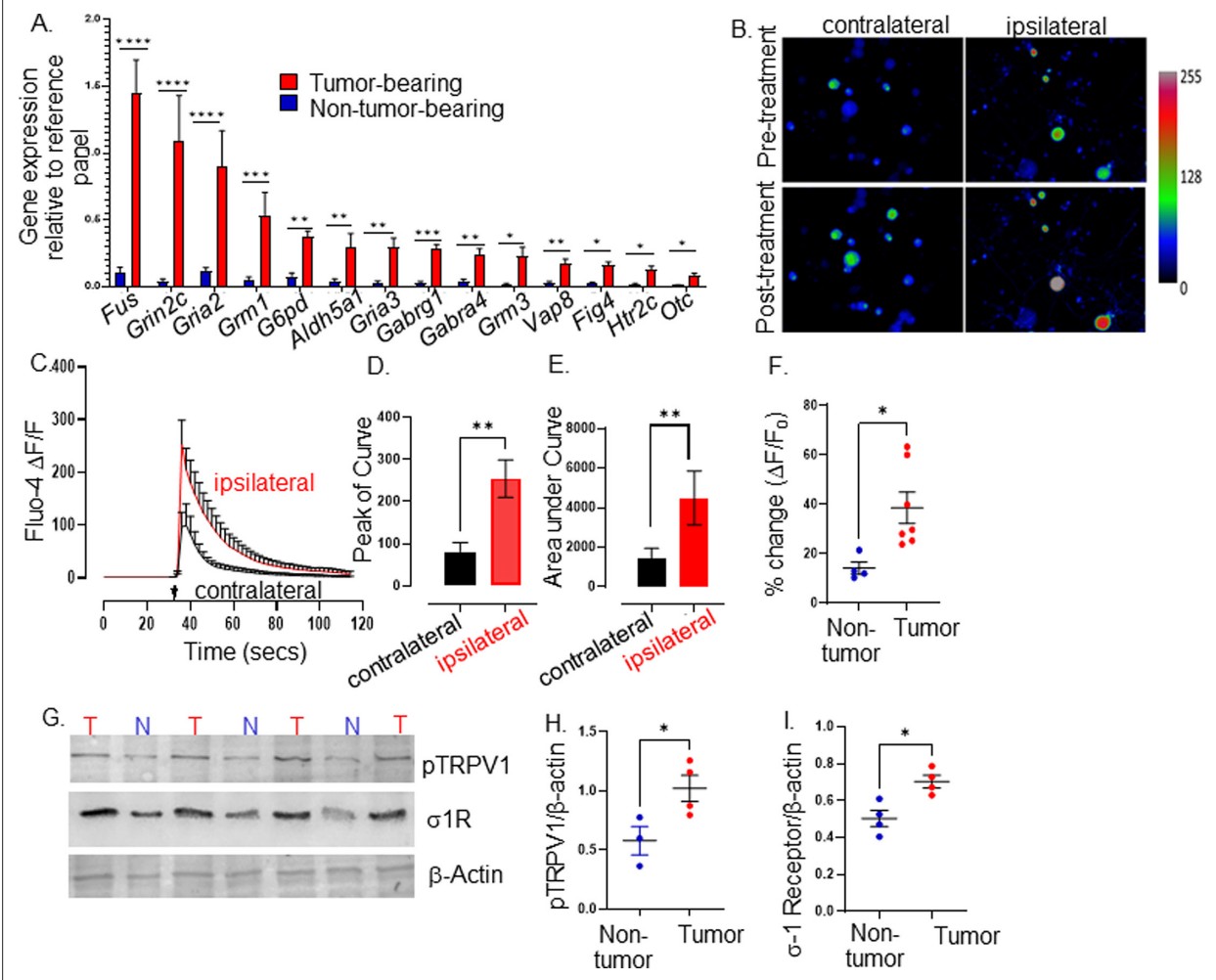

**Figure 2.** Tumor-infiltrating neurons become transcriptionally and functionally altered. Trigeminal (TGM) ganglia from tumor bearing (red) or non-tumor bearing (blue) mice were analyzed by (**A**) quantitative PCR array. N=4 TGM/group from n=4 mice/group; n=4 arrays/group. Ct values for each gene normalized to that of a housekeeping gene. Statistical analysis by multiple students t-test. *, p<0.05; **, p<0.01; ***, p<0.001, ****, p<0.00001. Data are expressed as means ± standard deviation. (**B**) Representative fluorescent images of dissociated neurons from the ipsilateral or contralateral TGM ganglia from a MOC2-7 tumor-bearing mouse. N=6 of each ganglia harvested from 6 tumor-bearing animals with n=4–8 neurons analyzed/ganglia. The color palette reflects the strength of the calcium signal. Images are taken pre- and post-treatment with capsaicin. (**C**) Graph of average change in fluorescence for all neurons analyzed in each group (ipsilateral and contralateral). Statistical analysis by student's t-test. Arrow, time of capsaicin (300 nM) stimulation. Graph of average peak of the curve (**D**) and area under the curve (**E**) from panel C. Statistical analysis by student's t-test. **, p<0.01. (**F**) Quantification of Ca$^{+2}$ responses to capsaicin (300 nM) from non-tumoral and tumor-infiltrating, tracer$^+$ (Tumor-infiltrating) neurons. N=4–8 neurons/group from n=4 control and n=7 tumor-bearing mice. Statistical analysis by paired students t-test, *, p<0.05. Data expressed as means ± SEM. (**G**) Representative western blot of trigeminal ganglia from non-tumor (**N**) or MOC2-7 tumor-bearing (**T**) animals for phosphorylated TRPV1 (pTRPV1, Ser502, Ser800), σ1 receptor (σ1R) and β-actin (loading control). Densitometric quantification of western blots for pTRPV1 (**H**) and σ1R (**I**). Statistical analysis by student's t-test. *, p<0.05. Data are expressed as means ± SEM. (n=3–4 mice/group). Blots were not stripped in between probing for different proteins.

The online version of this article includes the following source data for figure 2:

**Source data 1.** Uncropped and labeled gels for *Figure 2g*.

**Source data 2.** Uncropped and labeled gel for *Figure 2g*.

**Source data 3.** Raw unedited gel for *Figure 2g*.

**Source data 4.** Uncropped and labeled gel for *Figure 2g*.

**Source data 5.** Raw unedited gel for *Figure 2g*.

**Source data 6.** Uncropped and labeled gel for *Figure 2g*.

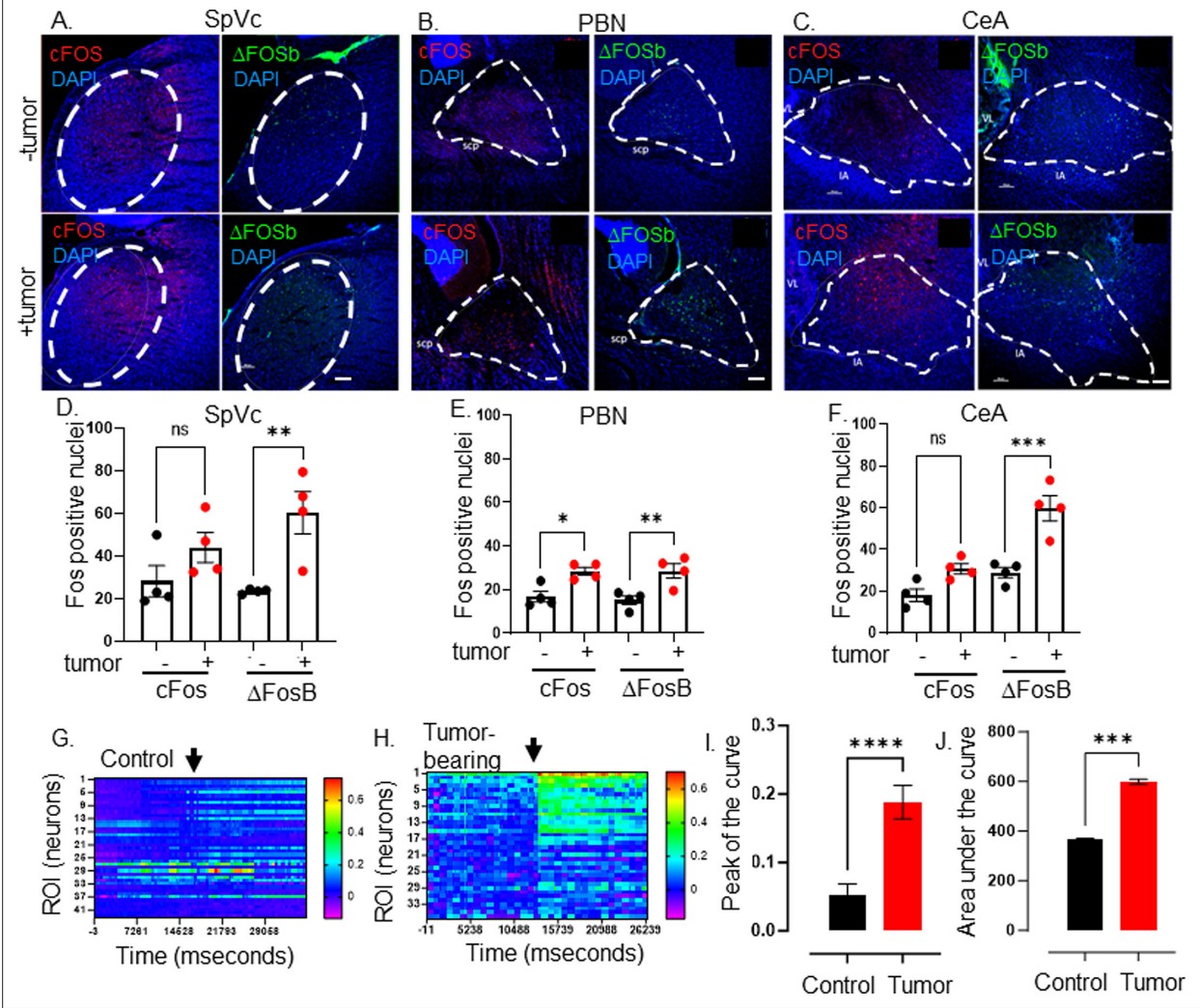

**Figure 3.** Elevated CNS neuronal activity in tumor bearing mice. Representative fluorescent photomicrographs of the spinal nucleus of the TGM (SpVc) (**A**), parabrachial nucleus (PBN) (**B**) and central amygdala (CeA) (**C**) from a control (-tumor) and MOC2-7 tumor-bearing (+tumor) mouse brains immuonfluorescently stained for cFos (red) or ΔFosB (green). Nuclei counterstained with DAPI (blue). Dotted circle denotes each brain region. Scale bar, 100 µm. Brain landmarks: Spc, superior cerebellar peduncle; VL, lateral ventricle; IA, intercalated amygdalar nucleus. Quantification of cFos and ΔFosB staining in SpVc (**D**), PBN (**E**) and CeA (**F**) from control (-tumor) and tumor-bearing (+tumor) mice. N=4 mice/group with n=2–4 sections analyzed/ brain region/mouse/group. Data are expressed as the mean ± SEM. Statistical analysis by one-way ANOVA*, p<0.05; **, p<0.01; ***, p<0.001; ns, not significant. Scale bar, 100 µm. Representative heat maps of regions of interest (ROI), each of which represent one neuron, within an ex vivo brain slice from non-tumor (control) (**G**) and tumor-bearing (**H**) mice. The color palette reflects the strength of the calcium signal. Arrow denotes the time of KCl (30 mM) application. Quantification of calcium imaging data (n=5 mice/group with n=2–4 sections analyzed/mouse) analyzing the peak of the curve (**I**) and the area under the curve (**J**). Statistical analysis by student's t-test. ****, p<0.0001; ***, p<001.

The online version of this article includes the following figure supplement(s) for figure 3:

**Figure supplement 1.** Neuronal activity.

**Figure supplement 2.** Neuronal activity.

**Figure supplement 3.** Neuronal activity.

## Tumor-brain circuit neurons harbor elevated activity

The transcriptional and functional changes evident in tracer[+] tumor-infiltrating neurons could result in alterations in central target neurons. To assess this, brain sections from MOC2-7 tumor-bearing and non-tumor-bearing animals were immunofluorescently stained for cFos and ΔFosB, two markers of neuronal activity with different courses of expression (*Chung, 2015*; *Hudson, 2018*; *Nestler et al., 2001*). ΔFosB expression was increased in several brain regions of tumor-bearing animals, while cFos

expression was also increased in the PBN (*Figure 3A–F*). Predominant differences in the long-lived ΔFosB were expected (*Nestler et al., 1999*), as those with the short-lived cFos changes are more challenging to capture using single time point assessment.

Next, we assessed the neuronal activity of tumor-bearing animal brains using stereotaxically injected AAV1-Syn-GCaMP6f, a viral vector encoding a neuron-specific synapsin-driven calcium sensor (*Chen et al., 2013b*), into the PBN. Two weeks after intra-cranial injection of the virus, mice were orally implanted with MOC2-7 cells. Approximately 18 days post-tumor inoculation, the animals were euthanized, and the neuronal calcium activity was recorded in ex vivo brain slices using a mini scope. While baseline fluorescence was similar between tumor-bearing and control animals, that recorded upon stimulation (KCl; 30 mM) was significantly higher in neurons of tumor-bearing animals (*Figure 3G, H, I and J*, *Figure 3—figure supplements 1–3*). These functional and Fos staining data indicate that central neurons within the tumor-brain circuit are functionally altered compared to their healthy brain counterparts.

## Ablation of tumor-infiltrating neurons attenuates cancer-induced brain alterations

Cancer patients, and even more notably, survivors, experienced poor mental health (*Carreira et al., 2018*; *Niedzwiedz et al., 2019*; *Utley et al., 2022*; *Ji et al., 2021*). The neural connection between tumor and brain, together with our finding that TRPV1-expressing nociceptor neurons within this circuit become functionally altered, might contribute to these changes. To test whether this is the case, we genetically engineered mice with ablated nociceptor neurons (TRPV1-Cre::Floxed-DTA). TRPV1-Cre::Floxed-DTA animals lack TRPV1-expressing cells (including neurons) as well as many other nociceptive neurons. These nociceptor-neuron-ablated mice have been previously characterized and lack the expected sensitivity to temperature as well as itch and pain reactions to chemical mediators such as capsaicin (*Mishra et al., 2011*). First, we confirmed the absence of TRPV1$^+$ neurons in the TGM of these ablated animals (*Figure 4A*). Fos immunostaining of brains from nociceptor ablated and control (C57BL/6) mice show no significant differences indicating that, in the absence of a malignancy, the neurons in these regions are not differentially activated (*Figure 4—figure supplements 1–3*). The absence of nociceptor neurons in tumor-bearing animals, however, decreased cFos and ΔFosB in the PBN, and ΔFosB in the SpVc (*Figure 4B and C*). Other tested regions were not impacted (*Figure 4D*). These data suggest that the tumor-brain communication was disrupted in the absence of nociceptor neurons.

## Alleviation of pain does not always restore behavior

To evaluate the effects of this disruption on cancer-induced behavioral changes, we assessed the animals' general well-being through nesting behavior (*Neely et al., 2019*) and anhedonia using the cookie test (*Liu et al., 2018*; *Eliwa et al., 2021*), as well as body weight and food disappearance as surrogates for oral pain and/or loss of appetite. Nociceptor-neuron-ablated mice showed increased nesting performance (*Figure 4E*) and decreased anhedonia (*Figure 4F*) compared to intact mice. This was accompanied by smaller tumor growth (*Figure 4G*) and increased survival (*Figure 4H*). Given the impact of nociceptor neuron ablation on tumor growth, we wondered whether differences in tumor volume contributed to the behavioral differences we noted. Thus, the behavior data were graphed as a function of tumor volume. A simple linear regression model was used to fit the data. In the case of nesting scores, the linear regression did not fit the data points very well making it difficult to assess nesting scores at a given tumor volume (*Figure 4—figure supplement 4*). However, the linear regression model fit the time to interact data better. Here, the data suggest that tumor volume did influence behavior as at any given tumor volume the time to interact with the cookie is generally smaller in TRPV1-Cre::Floxed-DTA animals as compared to C57BL/6 animals (*Figure 4—figure supplement 5*). While both groups showed similar body weight loss (*Figure 4I*), nociceptor ablated animals show a transient (Wk 1, 4, 5) decrease in food disappearance (*Figure 4J*). Significant effects for all statistical analyses in *Figure 4* are presented in *Supplementary file 2*.

The developmental ablation of TRPV1 neurons can lead to unintended effects on tumor innervation and behavior. Thus, we chemoablated TRPV1 neurons in 4-week-old juvenile C57BL/6 mice using resiniferatoxin (RTX) (*Liu et al., 2022a*; *Mishra and Hoon, 2010*). We examined nesting behavior in RTX and vehicle-treated C57BL/6 male mice orthotopically implanted with MOC2-7 cells. An additional

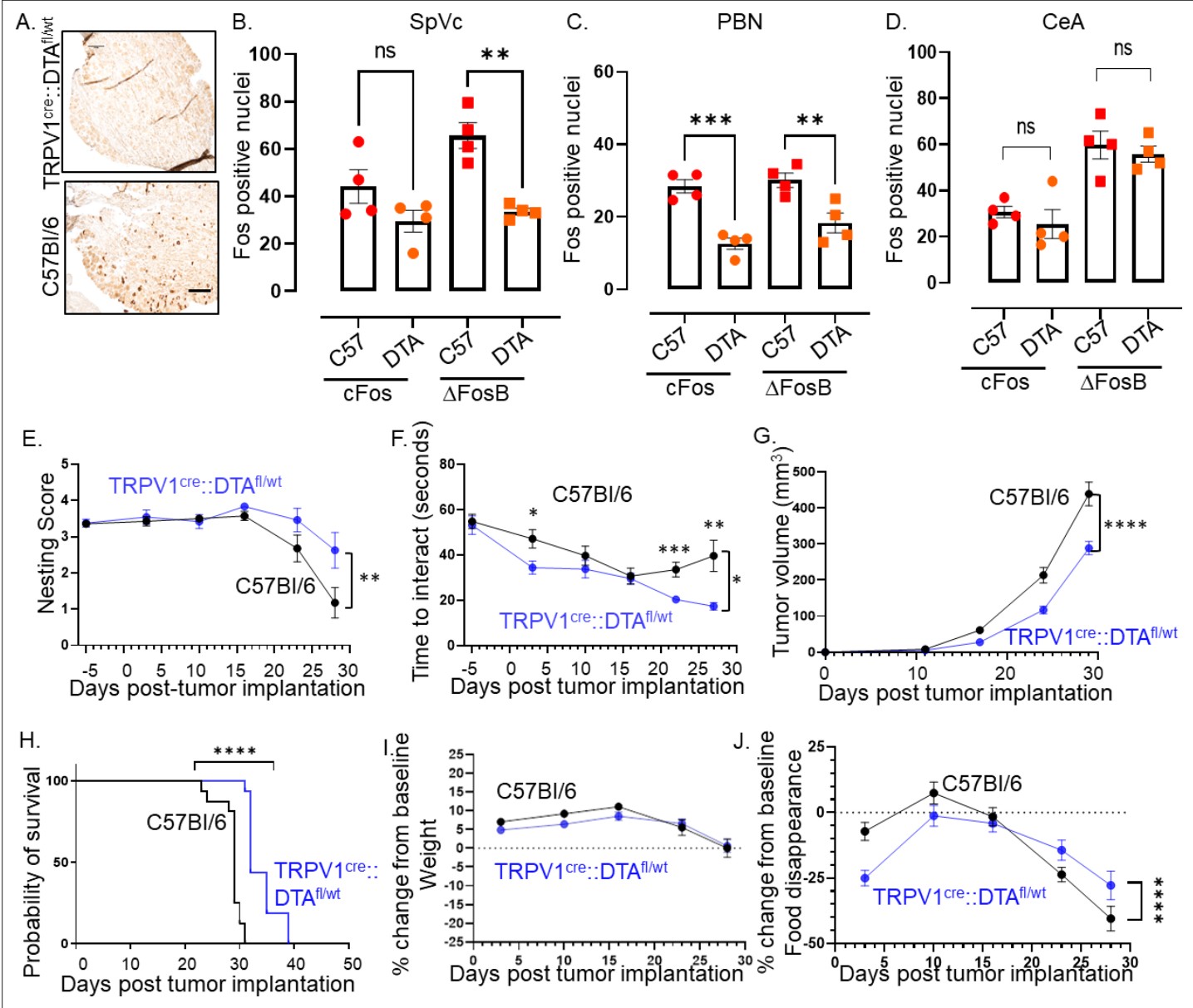

**Figure 4.** Intra-tumoral neurons impact behavior. (**A**) Bright field photomicrograph of TGM ganglion from C57BL/6 or TRPV1-Cre::Floxed-DTA mice IHC stained for TRPV1 (brown). Scale bar, 100 µm. Quantification of cFos and ΔFosB from the brains of C57BL/6 (**C57**) or TRPV1-Cre::Floxed-DTA (DTA) animals with MOC2-7 oral tumors in the Spinal nucleus of the trigeminal (SpVc) (**B**), the parabrachial nucleus (PBN) (**C**) and the central amygdala (CeA) (**D**). N=4 mice/group with n=2–4 sections analyzed/brain region/mouse. Statistical analysis by one-way ANOVA. **, p<0.01; ***, p<0.001. Data displayed as mean ± SEM. C57BL/6 (black, n=15) and TRPV1-Cre::Floxed-DTA (blue, n=14) mice were orthotopically implanted with MOC2-7 tumor and behavior assessed weekly and statistically analyzed by repeated measures ANOVA. (**E**) Graph of nesting scores over time. There is a main effect of time such that there is a decline in nesting performance over time. We also see a time by strain interaction in that C57BL/6 mice show a greater decline than the nociceptor neuron ablated TRPV1-Cre::Floxed-DTA mice. Post hoc testing shows a significant difference in nesting on day 27 (p=0.0315). (**F**) Graph of time to interact with the cookie in the cookie test. There is a main effect of time such that the time to interact with the cookie declines over time. There is a main effect of strain such that nociceptor neuron ablated TRPV1-Cre::Floxed-DTA mice were faster to interact with the cookie. Finally, there is a time by strain interaction such that nociceptor neuron ablated TRPV1-Cre::Floxed-DTA mice showed progressively faster task performance, while in C57BL/6 mice interaction time plateaued. Post hoc testing demonstrates a significant difference between the groups on days 3, 22, and 27 (p<0.05). (**G**) Tumor growth curves for mice that underwent behavioral testing. Statistical analysis by repeated measures ANOVA. There is a main effect of time such that there is an increase in tumor volume over time. We also see a time by strain interaction in that C57BL/6 mice have larger tumors than nociceptor ablated TRPV1-Cre::Floxed-DTA mice. (**H**) Kaplan-Meier survival curve for mice in panel I. Statistical analysis by Log-rank (Mantel-Cox) test. ****, p<0.0001. (**I**) Graph of % change in weight from baseline. Dotted line represents baseline. Statistical analysis by repeated measures ANOVA. There was a main effect of time such that both groups showed an initial increase in weight followed by a decrease. Post hoc testing shows a significant

*Figure 4 continued on next page*

*Figure 4 continued*

difference between the groups on days 5, 10, and 20. (**J**) Graph of % change from baseline in food consumption. Dotted line represents baseline. Statistical analysis by repeated measures ANOVA. There is a main effect of time such that there is a decline in % food disappearance over time. There is a time by strain interaction such that C57BL/6 mice show a greater decline in % food disappearance compared to nociceptor neuron ablated TRPV1-Cre::Floxed-DTA mice. In all panels, error bars are SEM.

The online version of this article includes the following figure supplement(s) for figure 4:

**Figure supplement 1.** Brain neuronal activity in controls.

**Figure supplement 2.** Brain neuronal activity in controls.

**Figure supplement 3.** Brain neuronal activity in controls.

**Figure supplement 4.** Tumor volume does not account for nesting differences.

**Figure supplement 5.** Tumor volume and cookie test.

control group consisted of age-matched C57BL/6 mice without tumors. Although all groups displayed similar nesting scores initially, the scores in vehicle-treated mice declined significantly faster compared to RTX-treated mice (***Figure 5A***). Body weight decreased in all tumor-bearing mice, but the extent of weight loss was similar between vehicle and RTX-treated mice (***Figure 5B***). Similar to the developmental ablation of TRPV1 neurons, their chemoablation post-development resulted in a similar decrease in tumor growth compared to controls (***Figure 5C***).

Since HNSCC tumors cause oral pain and both nesting and the cookie test require the use of the mouth, we also evaluated the effect of nociceptor ablation on voluntary wheel running to gain additional insights into cancer-associated fatigue, a surrogate for depressive-like behaviors in tumor-bearing mice (***Norden et al., 2015a***; ***Zombeck et al., 2013***; ***Grossberg et al., 2018***; ***Bower et al., 2011***; ***Norden et al., 2015b***). To assess the potential influence of cancer-associated pain, 1 week post tumor inoculation, groups of mice were treated with carprofen (a non-steroidal anti-inflammatory), extended-release buprenorphine (opioid), or vehicle. We found that wheel running decreased as tumors progressed, an effect partially alleviated by carprofen and buprenorphine (***Figure 6A and B***). The nesting behavior (***Figure 6C***), body weight (***Figure 6D***), and food disappearance (***Figure 6E***) were also alleviated by the painkillers. Neither carprofen nor buprenorphine impacted MOC2-7 tumor growth compared to vehicle-treated animals (***Figure 6F***), but buprenorphine increased tumor growth compared to carprofen-treated mice. Significant effects for all statistical analyses in ***Figure 6*** are presented in ***Supplementary file 3***. These findings suggest that choosing pain management drugs in the context of cancer can potentially influence tumor growth adversely. Furthermore, the data reveal that behaviors associated with the tumor site are adversely affected by cancer-associated pain. Pain-induced anhedonia is mediated by changes in the reward pathway. Specifically, in the context of pain, dopaminergic neurons in the ventral tegmental area (VTA) become less responsive to pain and

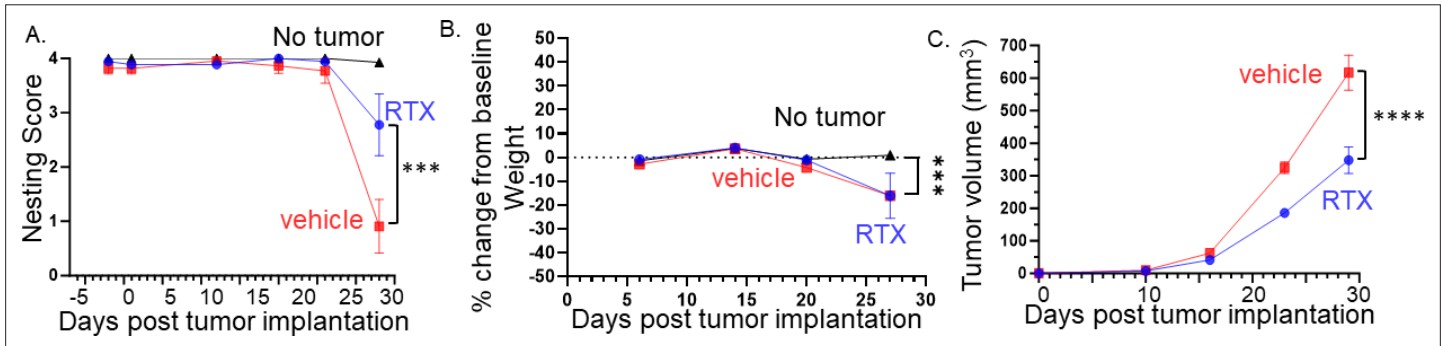

**Figure 5.** Denervation of tumors improves behavior, attenuates body weight loss, and reduces tumor growth. (**A**) Graph of nesting scores over time. Statistical analysis by repeated measures ANOVA (group x time): The time factor and its interaction with the group factor are significant with a significant difference between RTX- and vehicle-treated mice on the last time point (p<0.001). (**B**) Graph of % change in weight from baseline. Statistical analysis by repeated measures ANOVA (group x time): The time factor and its interaction with the group factor are significant with a significant difference between groups on the last time point (p<0.01). (**C**) Tumor volume according to treatment and time: Statistical analysis by repeated measures ANOVA (treatment x time): The time factor and its interaction with the group factor are significant with a significant difference between treatments on the last two time points (p<0.001). N=20 mice/group.

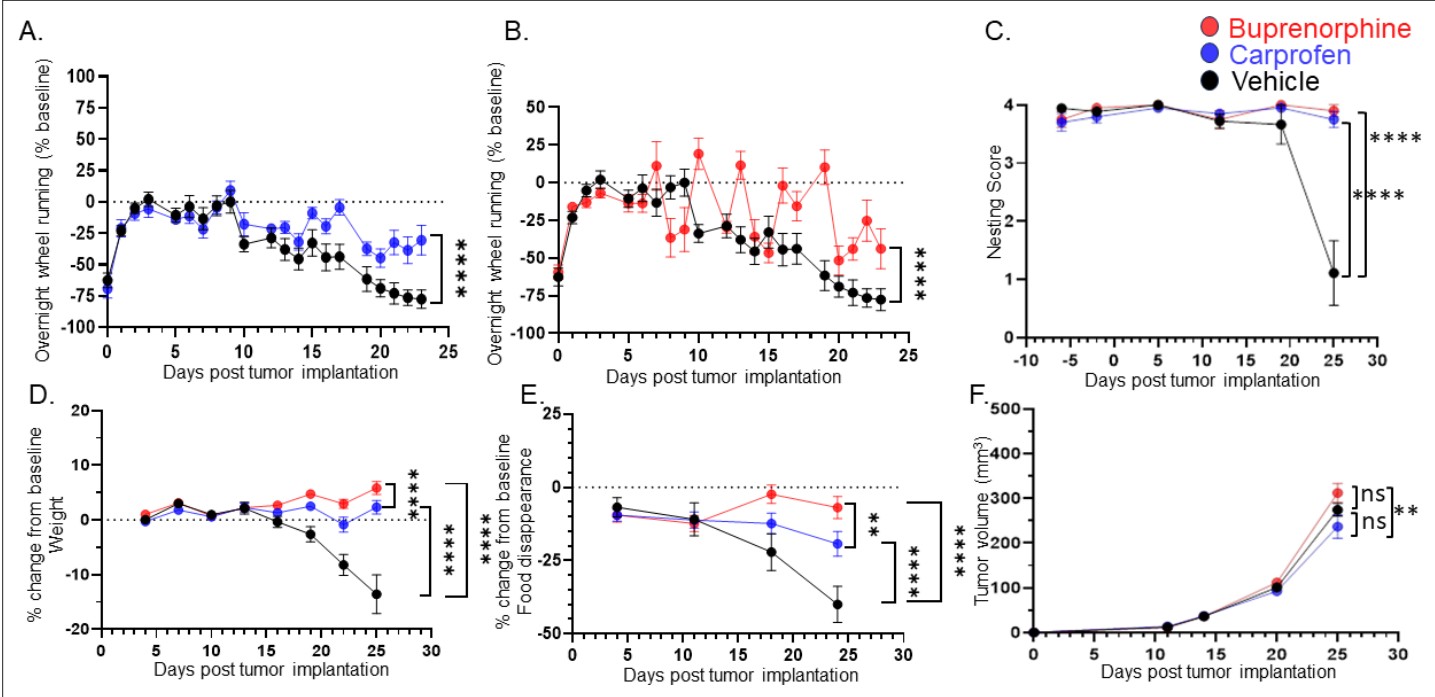

**Figure 6.** Treating for pain does not restore all behaviors. C57BL/6 male mice were orthotopically implanted with MOC2-7 tumors and separated into three groups: carprofen treated (10 mg/kg, blue, n=10), buprenorphine treated (3.25 mg/kg, red, n=10) or vehicle treated (black, n=9). Mice underwent behavioral testing. All analyses are by repeated measures ANOVA. (**A**) Data are graphed as % of baseline for overnight wheel running. The dotted line represents the baseline. There is a main effect of time such that there is a decline in voluntary wheel running over time. We also see a time by drug (carprofen) interaction in that vehicle-treated mice show a greater decline in wheel running than carprofen-treated mice. Post hoc testing shows a significant difference in wheel running (p<0.0001). (**B**) There is a main effect of time such that there is a decline in voluntary wheel running over time. We also see a time by drug (buprenorphine) interaction in that vehicle-treated mice show a greater decline in wheel running than the buprenorphine-treated mice. Post hoc testing shows a significant difference in wheel running (p<0.0001). (**C**) Graph of nesting scores over time. When comparing vehicle vs carprofen groups, there is a main effect of time such that there is a decline in nesting performance over time. We also see a time by drug (carprofen) interaction in that vehicle-treated mice show a greater decline than carprofen-treated mice. Similarly, we see a main effect of time when comparing the vehicle and buprenorphine groups. There is a time by drug (buprenorphine) interaction in that vehicle-treated mice show a greater decline than buprenorphine-treated mice. (**D**) Graph of % change in weight from baseline. The dotted line represents the baseline. There was a main effect of time. There is also a drug (carprofen) by time interaction in that vehicle-treated mice show a greater decline in weight than the carprofen-treated mice. In fact, carprofen-treated mice gain weight. Similarly, when comparing the vehicle and buprenorphine groups, there is a main effect of time. There is also an interaction of drug (buprenorphine) by time in that vehicle-treated mice show a greater decline in weight than the buprenorphine-treated mice. In fact, the buprenorphine-treated mice gain weight. When carprofen and buprenorphine-treated animals are compared, there is an interaction of time with treatment such that buprenorphine-treated animals gain more weight than carprofen-treated animals. (**E**) Graph of % change in food disappearance from baseline. When comparing the vehicle vs carprofen group, there is a main effect of time such that there is a decline in food disappearance over time. We also see a time by treatment interaction such that carprofen-treated animals do not decline in food disappearance as much as vehicle treated animals. When comparing the vehicle vs buprenorphine group, there is a main effect of time. In the buprenorphine treated group, there is a time by treatment interaction such that the buprenorphine-treated animals demonstrate the smallest decline in food disappearance. Finally, comparison of the carprofen and buprenorphine groups shows that there is an interaction between time and treatment such that buprenorphine-treated animals show the least reduction in food disappearance. (**F**) Tumor volume was monitored weekly for mice in all groups. There is a main effect of treatment such that mice treated with carprofen show a greater decline in tumor volume than buprenorphine-treated mice.

release less serotonin. This decreased serotonin results in disinhibition of GABA release; the resulting increased GABA promotes an increased inhibitory drive leading to anhedonia (*Markovic et al., 2021*) and, when extreme, anorexia. Carprofen and buprenorphine treatments completely reversed nesting behavior and significantly improved eating. Inflammation (*Bekhbat et al., 2022*) and opioids (*Le Merrer et al., 2009*) directly influence reward processing and though our tracing studies did not indicate that the tumor-brain circuit includes the VTA, this brain region may be indirectly impacted by tumor-induced pain in the oral cavity. Thus, an alternative interpretation of the data is that the effects of carprofen and buprenorphine treatments on nesting and food consumption may be due to inhibition of anhedonia (and anorexia) rather than, or in addition to, relieving oral pain (*Campos et al.,*

*2017*). However, merely alleviating pain (or inhibiting anhedonia) does not fully restore all behaviors, as evidenced by persistent issues in activities like wheel running.

## Discussion

We have previously demonstrated that head and neck squamous cell carcinoma is associated with innervation by sensory neurons and that substance P, one of the principal neuropeptides these fibers release, drives malignant cell proliferation and migration (*Restaino et al., 2023*). Significantly, patients suffering from HNSCC often experience depressive disorders, which are poorly managed clinically (*Chen et al., 2013a*; *Lazure et al., 2009*; *Elaldi et al., 2021*). Here, we found that nociceptor neurons infiltrating HNSCC connect to a pre-existing brain-projecting circuit (*Navratilova et al., 2016*; *Mercer Lindsay et al., 2021*) that includes the trigeminal ganglia, the spinal nucleus of the trigeminal (SpVc), as well as the parabrachial nucleus (PBN), and central amygdala (CeA) (*Ito et al., 2021*; *Jaramillo et al., 2021*; *Ge et al., 2022*; *Torres-Rodriguez et al., 2024*). This connection occurs independently of pain, and this cancer-brain circuit drives behavioral alterations in tumor-bearing mice. While the elimination of TRPV1-expressing nociceptor neurons reduces tumor growth and enhances survival rates, it also highlights a potential therapeutic target for mitigating depressive behaviors in cancer patients. This underscores the complex interplay between sensory neurons, cancer progression, and mental health, uncovering a two-pronged approach to improving HNSCC patients' physical and psychological health.

In melanoma (*Balood et al., 2022*) and HNSCC, we demonstrated significant alterations in transcript levels within tumor-infiltrating nerves. Several mechanisms can account for these changes. For instance, TRPV1 stimulation is sufficient to trigger the activation of the AP-1 transcription factor within neurons (*Backes et al., 2018*). Such TRPV1 activation could occur in response to low pH (*Boedtkjer and Pedersen, 2020*) or hypoxia (*Pocock and Hobert, 2008*; *Pocock and Hobert, 2010*; *Stevenson et al., 2012*), which are both prevalent within the tumor microenvironment. Alternatively, we previously demonstrated that tumor-released sEVs, which transport microRNAs (miRNAs), induced sprouting of loco-regional nerves to the tumor bed (*Madeo et al., 2018*; *Amit et al., 2020*). These sEV-transported miRNAs could also modulate various neuronal transcription factors and, in turn, drive the transcriptomic changes we observed in these neurons.

Our data indicate that these peripheral neuronal changes also influence the functioning of the brain areas to which they connect. Although we did not make the complete cartography of all brain regions used by tumor-infiltrating nerves, we identified a few critical nuclei. Subsequent work will use emerging circuit mapping techniques for whole brain profiling (*Ueda et al., 2020*). Nevertheless, our tracing in tumor-bearing nociceptor-neuron-ablated and intact animals was sufficient to functionally implicate TRPV1-expressing neurons in this communication.

Trigeminal ganglia neurons are composed of ~80% of $Na_V1.8^+$ nociceptor neurons, with one-half being peptidergic (TRPV1$^+$, TRPA1$^+$) neurons (*Kobayashi et al., 2005*; *Mickle et al., 2015*; *Mishra et al., 2011*). In the TRPV1-Cre::Floxed-DTA mouse, these peptidergic neurons are eliminated, leaving intact ~50% of pain-sensing neurons. The remaining presence of these non-peptidergic, largely MrgD$^+$, neurons might explain why nesting behavior in TRPV1-Cre::Floxed-DTA tumor-bearing animals is only partially restored.

Inflammation contributes to pain in cancer (*Renna et al., 2022*). Consistent with this, treatment of tumor-bearing mice with analgesic drugs (carprofen, buprenorphine) resulted in complete restoration of nest building and performance in the cookie test. These data show that the constraint imposed by the tumor on behavioral activities that require the use of the oral cavity is mainly due to pain and not to the physical interference of the tumor mass with the pattern of oral activities that need to take place for mice to perform the corresponding behavior.

Cancer pain often interferes with activities that don't involve oral movements, a phenomenon evident in the impaired voluntary wheel running observed in tumor-bearing mice. Although analgesic drugs can partially mitigate this deficit, they fail to restore this behavior fully, despite their efficacy in completely restoring oral activity behaviors like nesting and the cookie test. Moreover, given that carprofen and buprenorphine decrease inflammation (*Shepherd and Mohapatra, 2018*), their ability to restore normal nesting and cookie test behaviors (which require the use of the oral cavity where the tumor is located) suggests that inflammation at the tumor site contributed to the decline in these behaviors in vehicle-treated animals. Since both drugs were given systemically and each only partially

restored wheel running, it suggests that systemic inflammation alone cannot fully account for the decline in wheel running seen in vehicle-treated animals. We posit that the inflammation- and pain-independent component of this behavioral decline is mediated via the transcriptional and functional alterations in the cancer-brain circuit.

This suggests that factors beyond pain contribute to the observed behavioral changes. These alterations could be attributed to depression (*Walker et al., 2014*) or represent a competition between the energy demands of the HNSCC malignant cells and the host's skeletal muscles (*Grossberg et al., 2020*; *Thaker et al., 2006*; *Kim-Fuchs et al., 2014*).

Neuro-immune interactions have been studied in the context of a variety of conditions including, but not limited to infection (*Camp et al., 2021*), inflammation (*Kabata and Artis, 2019*; *Chen et al., 2020*), homeostasis in the gut (*van Baarle et al., 2023*; *Jacobson et al., 2021*; *Jakob et al., 2020*), as well as neurological diseases (*Tan et al., 2020*; *Jorfi et al., 2023*). Neuro-immune communications in the context of cancer and behavior have also been studied (e.g. sickness behavior, depression) (*Dantzer et al., 2008*; *Kopec et al., 2019*; *Dantzer, 2018*) however, these studies did not assess these interactions at the tumor bed. Investigations into neuro-immune interactions occurring within primary malignancies which harbor nerves have shed light on these critical communications. In the context of melanoma, which is innervated by sensory nerves, we identified that release of the neuropeptide calcitonin gene related peptide (CGRP) induces immune suppression. This effect is mediated by CGRP binding to its receptor, RAMP1, which is expressed on CD8 +T cells (*Balood et al., 2022*). A study utilizing a different syngeneic model of oral cancer similarly found an immune suppressive role for CGRP (*McIlvried et al., 2022*). These studies demonstrate that neuro-immune interactions occur at the tumor bed. Our current findings indicating that tumor-infiltrating nerves connect to a circuit that includes regions within the brain suggest that neuro-immune interactions within the peripheral malignancy may contribute to the behavioral alterations we studied.

Our current findings suggest that interrupting tumor-to-brain communication can mitigate the mental health decline often associated with cancer. Yet, a higher prevalence of depression persists among long-term cancer survivors compared to individuals without a cancer history. Having received treatment and showing no evidence of disease, these survivors continue to battle mental health issues (*Götze et al., 2020*; *Kuba et al., 2019*). This phenomenon prompts a critical question - why does this discrepancy exist? It is conceivable that these patients, through treatments like surgery, radiation, or chemotherapy, have disrupted the tumor-brain connection and have sustained this disconnection over the years. However, unlike animal models, their mental health decline is not reversed. Several speculations, grounded in known factors, can be considered.

The human experience with cancer is distinct from that of mouse models; individuals can live with malignant growths for extended periods, often unaware of their existence due to asymptomatic or nonspecific symptoms. During this undiagnosed period, the tumor-brain circuit might induce irreversible changes in central neurons through activity-dependent transcriptional modifications (*Yap and Greenberg, 2018*; *Takayanagi-Kiya and Kiya, 2019*). These alterations, anchored in neuronal plasticity, adaptation, and behavior, may become permanent even after the cancer is diagnosed and treated.

Moving forward, to reverse these entrenched brain changes, understanding the specific transcriptional alterations incurred by central neurons is vital. Unraveling this complexity could pave the way for therapies that reverse the neuronal activity impacts, fostering improved mental health. Although in its early stages, our insight into the role of nerves in cancer is growing and highlighting neuronal targets for pharmacological interventions.

Our study is not exempt from limitations. First, our exploration of the tumor-to-brain pathway was confined to mice with HNSCC and used a single cancer cell line. Hence, the universality of our conclusions warrants verification through additional HNSCC cell lines and diverse cancer models. Furthermore, the exclusive focus on HNSCC, particularly considering the nerve-rich oral region, raises questions about the pathway's establishment and behavioral impact in other contexts. Our behavioral assessment was not exhaustive; a comprehensive analysis of nerve-dependent and independent behaviors remains forthcoming. Behaviors could also be influenced by elements like soluble factors or the energy competition between the tumor and behavioral activities.

Head and neck cancer is predominantly a cancer in males; it occurs in males three times more often than in females (*Gaeta et al., 2023*), this disparity increases in certain parts of the world.

While smoking cigarettes and drinking alcohol are risk factors for HPV negative head and neck squamous cell carcinoma, even males that do not smoke and drink are have a higher susceptibility for this cancer than females (*Park et al., 2022*; *Dittberner et al., 2020*). Thus, our studies used only male mice. However, we do recognize that females also get this cancer. In fact, female patients with head and neck cancer, particularly oral cancer, report more pain than their male counterparts (*Bianchini et al., 2019*; *Reyes-Gibby et al., 2014*). These findings suggest that differences in tumor innervation exist in males and females. Our studies have also solely involved male mice, presenting a clear gap in understanding the potential sex differences in the development of the tumor-to-brain communication pathway and its subsequent influence on behavior. The inclusion of female subjects in future research is essential to provide a comprehensive insight into these processes.

Second, our nerve tracing methodology, where we injected 2 μl of WGA into the tumor bed, is designed to label only the tumor-infiltrating nerves and their connections. However, this approach fails to label *all* tumor-infiltrating nerves, predominantly those not close to the WGA-injected region and nerves that are in contact with or influenced by the tumor but are not directly infiltrating it. This limitation is a calculated one, reflecting a technical trade-off. We aimed to ensure the specificity of capturing only tumor-infiltrating nerves, which meant sacrificing the comprehensiveness of labeling all nerves associated with the tumor. This constraint extends to our calcium imaging studies and brain tracing. Although we can confidently assert that all labeled neurons are indeed tumor-infiltrating or connected, we cannot conclusively state that all tracer-negative ipsilateral nerves are unconnected to the tumor-brain circuit.

Even with these constraints, our findings are important. We have demonstrated that tumor-infiltrating nerves are integrated into a pre-established neuronal circuit. This circuit extends from the tumor bed to the TGM ganglion, connects to the SpVc, and projects into the brain, influencing behavior in both pain-dependent and independent ways. This revelation paves the way for in-depth exploration into the mechanistic underpinnings of cancer-associated alterations in well-being and mental health, shedding light on potential therapeutic interventions to alleviate these profound effects.

## Materials and methods

### Study approval

All animal studies were performed with approval from the Institutional Animal Care and Use Committee at Sanford Research and were within institutional guidelines and complied with all relevant ethical regulations. Sanford Research has an Animal Welfare Assurance on file with the Office of Laboratory Animal Welfare (assurance number: A-4568–01) and is accredited by AAALAC, Intl. Sanford Health is also a licensed research facility under the authority of the United States Department of Agriculture (USDA) with USDA certificate number 46 R-011.

### Inclusion and ethics statement

All studies utilizing animals were performed with approval from the appropriate ethical bodies to ensure sufficient protection was in place (Sanford IACUC protocol# 2023–0105). This study was a collaborative, multi-disciplinary, multi-institutional effort with contributions from researchers in academic positions.

### Cell lines

C57BL/6-derived MOC2-7 cells (RRID:CVCL_ZD34; *Judd et al., 2012* previously known as MOC7) (*Onken et al., 2014*) were a kind gift from Dr. Ravindra Uppaluri (Dana-Farber Cancer Institute, Boston, MA). They were authenticated by STR profiling (ATCC Cell Line Authentication Service) and tested as mycoplasma negative by a Sanford Research Core facility. They were maintained in DMEM with 10% fetal calf serum and cultured at 37 °C and 5% $CO_2$, with culture medium refreshed every 3 days. MOC2-7 tumors have been characterized as non-inflamed and poorly immunogenic (*Moore et al., 2016b*; *Moore et al., 2016a*; *Shah et al., 2016*). These cells can be obtained through Kerafest (https://www.kerafest.com).

## Animal studies

All animal experiments were performed in the Sanford Research Animal Resource Center which is a specific pathogen-free facility. All mice were maintained in IVC Tecniplast Green line Seal Safe Plus cages which were opened only under aseptic conditions in an animal transfer station. All cages were changed every other week using aseptic technique. All cages had individual HEPA filtered air. Animal rooms were maintained at 75 °F, 30–70% humidity, with a minimum of 15 air changes per hour, and a 14:10 light/dark cycle. Corncob bedding, which was autoclaved prior to use, was maintained in all cages. Irradiated, sterile food (Envigo) and acidified water (pH 2.8–3.0) were available ad libitum. There was a maximum of 5 mice/cage. All animals were observed daily for abnormal behavior, signs of illness or distress, the availability of food and water and proper husbandry. Animals injected with murine tumor cells were 10-week-old C57BL/6 or TRPV1-Cre::Floxed-DTA mice (The Jackson Laboratory) weighing approximately 24 g. Investigators were blinded to the groups when assessing animals (e.g. measuring tumors). Animals were numbered by ear punch and cage number. For all studies, animals were randomized into groups.

Nociceptor-neuron-ablated TRPV1-Cre::Floxed-DTA animals were generated by crossing ROSA26-DTA (diphtheria toxin A, B6.129P2-*Gt(ROSA)26Sor*<sup>tm1(DTA)Lky</sup>/J)(Jax#009669) and TRPV1-Cre (B6.129-*Trpv1*<sup>tm1(cre)Bbm</sup>/J) (*Cavanaugh et al., 2011*) (Jax# 017769) mice; the progeny (TRPV1-Cre::Floxed-DTA) express DTA (diphtheria toxin fragment A) under control of the TRPV1 promoter, thereby genetically ablating all TRPV1 expressing cells (including TRPV1-expressing neurons) throughout development. Absence of TRPV1-expressing sensory neurons was validated by IHC staining of TGM ganglia for TRPV1.

Chemoablation of TRPV1-expressing neurons post-development was accomplished with resiniferatoxin (RTX), a naturally occurring potent TRPV1 agonist. RTX treatment results in the specific ablation of TRPV1 neurons (post-development), while sparing other TRPV1 expressing (non-neuronal) cells. RTX treatment in mice consisted of three injections of RTX given 24 hr apart. The injections are given in the flank and were: 30 µg/kg on day 1, 70 µg/kg on day 2, and 100 µg/kg on day 3. Just prior to the first RTX injection, mice are treated with 0.1 mg/kg of buprenorphine, given intraperitoneally, as the first injection causes pain. Subsequent injections do not require buprenorphine. It takes 4 weeks for TRPV1 nerve ablation to occur. Control mice received injections with vehicle solution (DMSO with Tween80 in PBS). The tail flick test was used to confirm nerve ablation.

## Tail flick test

Prior to testing, mice are acclimated to the room for 30 min. A water bath is pre-warmed to 52 °C and a thermometer is used to constantly monitor the temperature of the water. The mice are picked up using a fresh paper towel so that only their tail is outside the paper towel. Once the mouse is relaxed in the tester's hand, the mouse's tail is lowered into the water so that a quarter of the tail is submerged. The time for the tail to be flicked out of the water is recorded. If no response has occurred by 20 s, the mouse's tail is removed from the water to prevent tissue damage. A response time latency greater than 10 s signifies denervation.

## Tumor implantation

The cells injected into animals were cultured in media containing 10% fetal calf serum. When cells are harvested for tumor injections, they are first washed two times with PBS and then trypsinized to detach the cells from the plate. Cells are collected, washed again with PBS and resuspended with DMEM without serum; this is what is injected into animals. We harvest cells in this way in order to eliminate any serum being injected into mice. Ketamine (87.5 mg/kg)/xylazine (10 mg/kg) were used to anesthetize mice prior to oral cavity tumor implantation. Tumors were initiated as follows: using a 25-gauge needle, cells ($5 \times 10^4$ cells in a total of 50 µl) were implanted orthotopically in the oral cavity of male C57BL/6 mice. Specifically, the needle was placed proximal to the crease of the mouse cheek, inserted at approximately a 45° angle, bevel deep, and tumor cells injected into this location of the cheek pouch. Mice were allowed to recover from anesthesia on a heating blanket and returned to their home cage. Control (non-tumor) mice were treated the same way only their injection contained media alone (no cells). Tumor growth was monitored weekly by caliper measurements of isoflurane anesthetized animals and tumor volume calculated was using the formula: ½ (length*(width^2)).

Non-tumor control mice were also anesthetized with isoflurane each time that tumor-bearing animals were anesthetized for tumor measurements.

## Euthanasia criteria

Criteria for euthanasia in our IACUC approved protocol include maximum tumor volume of 1000 mm$^3$, edema, extended period of weight loss progressing to emaciation, impaired mobility or lesions interfering with eating, drinking or ambulation, rapid weight loss (>20% in 1 week), as well as weight loss at or more than 20% of baseline. In addition to tumor size and weight loss, we use the body condition score to evaluate the state of animals and to determine euthanasia.

## Drug treatments

Carprofen (Pivetal) was provided in the drinking water (10 mg/kg which is equivalent to 0.067 mg/ml) which was refreshed every 7 days. Buprenorphine-ER (Fidelis Animal Health), a 72 hr extended-release formulation, was given by subcutaneous injection (3.25 mg/kg) every 72 hr. Vehicle-treated animals were treated with subcutaneous injection of saline. Drug treatments commenced on day 7 post-tumor implantation in all groups. Given that buprenorphine treatment requires a subcutaneous injection every 3 days, mice in the other groups (vehicle and carprofen) received a subcutaneous injection of saline at the same volume (60 µl) as the buprenorphine-treated animals. In this way, all animal stress from handling and injection was the same.

## Nest building

Nest building is an innate behavior performed in rodents of both sexes and is a general indication of well-being (*Neely et al., 2019*); it also measures the motivation to perform a goal-directed task. To assess nesting behavior, individually housed mice were given a nesting square (Ancare) overnight. The following morning, nests were scored by two independent scorers who were blinded to the conditions (*Deacon, 2006*; *Kraeuter et al., 1916*). The scoring system is 0–4 and based on the percentage of the nestlet that is shredded and the height of the nest. Two baseline measurements were conducted prior to tumor implantation followed by weekly testing. Exclusion criteria of an average baseline score of 3 or greater was used to ensure all mice were good nest builders at baseline (n=3 C57BL/6 mice and n=7 TRPV1-Cre::Floxed-DTA mice were excluded for this reason).

## Cookie test

The cookie test is a variation of the sucrose preference test, a test of anhedonia (the inability to experience enjoyment or interest in previously rewarding activities). This test assesses the animal's endogenous inclination for sweet tastes and the notion that it derives pleasure from consuming sweets. One hour prior to testing, mouse food is removed and the animals are acclimated to the brightly lit testing room. Following the acclimation period, mice were given a piece of cookie (approximately 1.5 g). The time it took the mouse to bite the cookie from its placement in the cage was measured to determine time-to-interact. Each mouse was acclimated to the cookie every other day for 2 weeks prior to tumor implantation with a baseline measurement followed by weekly testing. Mice were individually housed. One mouse from the C57BL/6 group was excluded due to incomplete data (mouse died before completion of the study).

## Voluntary wheel running

Prior to tumor implantation, mice were singly housed with a running wheel maintained in the home cage continuously. The only time that the wheel was removed was when mice were undergoing nesting. The mice were acclimated to the wheel for 2 weeks prior to tumor implantation. This acclimatization period provided the time to stabilize their running performance. Running wheels were maintained in the home cage for the duration of the study. As mice are most active during the dark phase, nighttime running (8pm to 8am) was assessed. Running data were collected in 1 min bins continuously throughout the duration of the experiment.

## Food disappearance

Animals were provided 100 g of solid food per cage each week. At the end of each week, the remaining food was weighed, and this value was used to calculate the amount of food that disappeared in that week per cage. For all behavioral experiments, mice were singly housed.

## Neural tracing of orthotopic HNSCC tumors

When oral tumors reached approximately 5x5 mm in size, neural tracer was injected into the tumor as described below. Ketamine (87.5 mg/kg)/xylazine (10 mg/kg) were used to anesthetize the mice. A 10 µL Hamilton syringe with 30 G needle was loaded with 1% Wheat Germ Agglutinin (WGA) conjugated to either AlexaFluor 594 (Invitrogen) or CF568 (Biotium) in PBS and 2 µl was slowly injected intra-tumorally with the bevel side up. The needle was inserted approximately midway through the tumor then pulled back slightly to reduce pressure and leakage of tracer following injection. The injection of tracer occurred slowly over the course of 10 min. After the tracer was injected, the needle was kept in place for 2 additional minutes before being slowly removed. Mice were placed on a heating pad until recovery from anesthesia. Five to 7 days later, mice were deeply anesthetized and transcardially perfused with ice cold PBS followed by 4% paraformaldehyde and trigeminal ganglia (TGM) were carefully removed and placed in HBSS in a 24-well plate kept on ice. TGM harvesting was completed as follows: euthanized animals were subjected to a midline incision while in the prone position to expose the crown of the skull. The brainstem was separated from the spinal cord by a transverse cut and the top of the skull was removed, exposing the brainstem and TGM. All tissues (tumor, ganglia and brain) were fixed, sectioned, and imaged for WGA labeled nerve fibers and somas under confocal microscopy. WGA injection into control (non-tumor bearing animals) was performed the same way and injection was into the same oral region where tumors were implanted. Volumes of 2 or 10 µl of WGA were used.

## Immunohistochemical (IHC) staining

Tissues were formalin fixed, paraffin-embedded and cut into 5 µm sections. The BenchMark XT automated slide staining system (Ventana Medical Systems, Inc) was used to optimize antibody dilutions and staining. The Ventana CC1 solution was used to perform the antigen retrieval step (basic pH tris base buffer). Tissues were incubated in primary antibody for 1 hr. The Ventana iView DAB detection kit was used as the chromogen and the slides were counterstained with hematoxylin.

## Antibody used for immunohistochemistry (IHC)

Anti-β-III Tubulin (Abcam, Cat# ab78078, 1:250, RRID: AB_2256751), Anti-TRPV1 (Alomone Labs Cat# ACC-030, 1:400, RRID:AB_2313819).

## Antibodies used for immunofluorescence (IF)

cFos (Cell Signaling, Cat# 2250, 1:10,000, RRID: AB_2247211), ΔFosB (Abcam, Ab11959, 1:5000, RRID:AB_298732). The following secondary antibodies were used: Alexa568 anti-rabbit (Thermo Fisher, Cat# A-11011, 1:500, RRID:AB_143157), Alexa488 anti-mouse (Thermo Fisher, Cat# A-11001, 1:500, RRID:AB_2534069).

## Antibodies used for western blot

β-actin (Sigma Life Science, Cat#A2228, 1:1000, RRID: AB_476697), Tau (Abcam, Cat# ab75714, 1:500, RRID:AB_1310734), phosphorylated TRPV1 (Thermo Fisher Scientific, Cat# PA5-64860, 1:500, RRID:AB_2663797), Sigma-1R (ProteinTech, Cat# 15168–1-AP, 1:1000, RRID:AB_2301712), Doublecortin (DCX) (Santa cruz, Cat# sc-271390, 1:1000, RRID:AB_10610966).

## Brain immunostaining and analysis

Brains were post-fixed for 24 hr. Coronal or sagittal sections were cut at 40 µm using a vibratome. Sequential fluorescent immunolabeling was performed for cFos or ΔFosB on free-floating sections. All sections were blocked with 10% goat serum in 0.3% Tx-100. All antibodies were diluted in PBS containing 0.03% Triton X-100 and 2% normal goat serum. Sections were incubated in Fos antibody for 72 hr. Following washes in 0.1 M phosphate buffered saline, sections were incubated in secondary antibody for 3 hr. Following additional washes, sections were mounted onto glass slides, and cover

slipped with cytoseal prior to imaging. The total numbers of Fos-labeled nuclei were counted on 10×2 D images acquired using a laser-scanning confocal microscope (Nikon A1R) and verified as positive if the signal filled the nucleus and stood out clearly compared to surrounding tissue *Kee et al., 2007*. Nuclei were counterstained with DAPI. Quantification of Fos-labeled cells was performed using digital thresholding of Fos-immunoreactive nuclei. The threshold for detection was set at a level where dark Fos-immunoreactive nuclei were counted, but nuclei with light labeling, similar to background staining, were not. For each animal, 3–4 sections were selected at 120 µm intervals. ImageJ cell counter (v. 1.52) was used to quantify Fos +nuclei per region / section.

## Analysis of transcriptional changes in tumor-infiltrating neurons

To assess transcriptional changes in tumor-infiltrating neurons we utilize a qPCR array (ScienCell, GeneQuery Neural Transmission and Membrane Trafficking, #MGK008). These qPCR-ready 96 well plates enable rapid profiling of 88 key genes important for neuronal functions (listed along with built-in control genes in *Supplementary file 1*). Mice bearing MOC2-7 oral tumors were injected with tracer (as described above); TGM ganglia were isolated on day 28 post-tumor implantation, their RNA harvested, converted into cDNA and then assayed with the array as per manufacturer's instructions (ScienCell, GeneQuery arrays). We used N=4 TGM/group from n=4 mice/group; n=4 plates/condition. Control RNA was isolated from TGM ganglia from age-matched non-tumor bearing mice (n=4 mice). Relative gene expression was calculated as per manufacturer's recommendations.

## Western blot analysis of whole tumor lysate

Tumors were excised from euthanized mice, taking care to eliminate as much non-tumor tissue as possible. Tumors were then placed in approximately 500 µl lysis buffer (50 mM Tris HCl pH 7.4, 100 mM NaCl, 100 mM NaF, 10 mM NaPPi, 2 mM $Na_3VO_4$, 10% glycerol, HALT protease inhibitor cocktail) with 1% TX-100 and kept on ice for 10 min. The homogenate was sonicated 3 x for 15 s and then incubated on ice for 15 min. Samples were then centrifuged at 2000 × $g$ for 5 min at 4 °C, supernatant was collected and further centrifuged at 12,000 × $g$ for 10 min at 4 °C. Protein concentrations of the supernatants were determined by BCA protein assay.

## Western blot analysis of ganglia lysate

Trigeminal ganglia were harvested as described above. Ganglia were lysed using a homogenizer in lysis buffer at 4 °C. Homogenates were centrifuged at 10,000 × $g$ for 20 min and protein concentrations determined from the supernatant. Protein concentration was measured by BCA protein assay (Pierce, Cat#23225) and 40 µg of total protein were separated by SDS-PAGE, transferred to PDVF membranes which were then blocked for 30 min at room temperature (RT) in 5% milk in PBS. Membranes were incubated with primary antibody overnight at 4 °C. Following 1xTBST washes, membranes were probed with an HRP-conjugated secondary antibody (1:10,000 dilution) for 1 hr at RT, washed and imaged on a Li-COR Odyssey FC imaging system. Densitometric quantification was used to assess changes in protein expression.

## Western blot densitometry analysis

Raw images of western blots were analyzed using ImageJ. Briefly, images were opened and bands of interested were selected by gating with the rectangle selection tool. Densitometry was measured based on grey scale analysis of selected area. Analysis was conducted on proteins of interest as well as the loading control band (β-actin). Relative expression of proteins of interest was determined as a fraction of β-actin densitometry.

## Neuron culture

Trigeminal ganglia were harvested from MOC2-7 tumor-bearing or non-tumor animals and enzyme digested in papain, and then collagenase II/dispase at 37°C for 15 min. After washing and trituration, cells were plated onto a thin layer of Matrigel in glass bottom dishes and cultured with HamsF12 supplemented with 10% FBS. The cells were maintained in an incubator (5% $CO_2$, 37 °C) for 24 hr before they were used for calcium imaging experiments.

## Ca2+ imaging

TGM neurons from non-tumor and tumor-bearing animals (n=4–6 mice/condition) were imaged on the same day. Neurons were incubated with the calcium indicator, Fluo-4AM, at 37 °C for 20 min. After dye loading, the cells were washed, and Live Cell Imaging Solution (Thermo-Fisher) with 20 mM glucose was added. Calcium imaging was conducted at room temperature. Changes in intracellular $Ca^{2+}$ were measured using a Nikon scanning confocal microscope with a 10 x objective. Fluo-4AM was excited at 488 nm using an argon laser with intensity attenuated to 1%. The fluorescence images were acquired in the confocal frame (1024 × 1024 pixels) scan mode. After 1 min of baseline measurement, capsaicin (300 nM final concentration) was added. $Ca^{2+}$ images were recorded before, during and after capsaicin application. Image acquisition and analysis were achieved using NIS-Elements imaging software. Fluo-4AM responses were standardized and shown as percent change from the initial frame. Data are presented as the relative change in fluorescence ($\Delta F/F_0$), where $F_0$ is the basal fluorescence and $\Delta F = F \, F_0$ with F being the measured intensity recorded during the experiment. Calcium responses were analyzed only for neurons responding to ionomycin (10 μM, positive control) to ensure neuronal health. Treatment with the cell permeable $Ca^{2+}$ chelator, BAPTA (200 μM), served as a negative control.

## Stereotaxic AAV injection

Male mice were anesthetized with 2% isoflurane and placed in a stereotaxic head frame on a heating pad. A midline incision was made down the scalp and a craniotomy was made using a micro drill. A 10 μl Hamilton syringe was used to infuse 1 μl of AAV1/Syn-GCaMP6f-WPRESV40 (titer $4.65 \times 10^{13}$ GC per ml, via Addgene) into the parabrachial nucleus (−5.3 mm anteroposterior, −1.3 mm mediolateral, −3 mm dorsoventral) via a microsyringe pump. After infusion, the needle was kept at the injection site for 5 min and then slowly withdrawn. Two weeks following stereotaxic surgeries, tumors were introduced by implanting MOC2-7 cells orthotopically into the oral cavity as described above.

## Ex vivo Ca2+ imaging of brain slices

Two weeks post-tumor (or sham) implantation, wildtype and tumor-bearing mice (n=3 mice/group) were anesthetized under isoflurane and perfused intracardially with 10 ml of ice-cold N-methyl-d-glucamine (NMDG) solution (92 mM NMDG, 30 mM $NaHCO_3$, 25 mM glucose, 20 mM Hepes, 10 mM $MgSO_4$, 5 mM sodium ascorbate, 3 mM sodium pyruvate, 2.5 mM KCl, 2 mM thiourea, 1.25 mM $NaH_2PO_4$, and 0.5 mM $CaCl_2$ [pH 7.3, 300 mOsm, bubbled with 95% O2 and 5% $CO_2$]) (*Ting et al., 2018*). The brains were quickly removed and placed into additional ice-cold NMDG solution for slicing. Coronal slices (150 μm) were cut using a vibratome (n=3 slices/brain). Slices were transferred to Hepes holding solution and warmed to 37 °C (bubbled with 95% $O_2$ and 5% $CO_2$) for 1 hour. After incubation, slices were transferred to the recording chamber with RT (22° to 25 °C) recording solution. A miniaturized microscope (Miniscope V4) imaged the GCaMP6f signal from the slices. Each video was processed with motion correction and ΔF/F calculation. Regions of interest (ROIs; considered as a single-cell soma) were manually selected. ROIs that exhibited short bursts of ΔF/F changes or fluctuations during the recording were analyzed. The ΔF/F changes were then aligned with the time window of treatment. Movement correction was performed using the motion correction module in the EZcalcium toolbox employed in MATLAB *Cantu et al., 2020*. The fluorescence intensity trace of each neuron was extracted, and ΔF/F was calculated (CNMF).

## Statistical analysis

GraphPad Prism (version 10.0.3, 2023) was used for all statistical analyses.

## Gene expression

For qRT-PCR analysis, Ct values for each gene were normalized to that of the reference gene. Statistical analysis by multiple student's t-test.

## Ca2+ imaging

Statistical analysis by unpaired student's t-test. Fluo-4AM/GCaMP6f responses were standardized and shown as percent change from the initial frame. Data are presented as the relative change in fluorescence ($\Delta F/F_0$), where $F_0$ is the basal fluorescence and $\Delta F = F \, F_0$ with F being the peak response.

### Western blot

Membranes were visualized, and proteins were quantified using the Odyssey infrared imaging system and software (Li-COR). Densitometric quantification of western blots was performed by normalizing the signal from tumor or non-tumor ganglia to the β-actin (loading control) signal and differences assessed by one-way ANOVA with post-hoc Tukey test.

### Fos brain immunostaining

The numbers of Fos (cFos, ΔFosB) immune-positive neurons in brain sections from tumor-bearing or non-tumor animals were quantified as described above and statistically analyzed by two-way ANOVA with multiple comparisons.

### Behavior Assays

The variations over time of the nesting scores for each group were statistically analyzed by repeated measures ANOVA. Exclusion criteria of an average baseline score of 3 or greater was used to ensure all mice were good nest builders.

Time to interact scores (cookie test): Statistical analysis by repeated measures ANOVA.

Voluntary wheel running: The data were collected in 1 min bins continuously. When compiling the data, the sum of bins between 8pm to 8am is taken for each individual animal for voluntary nightly running. Baseline is calculated by taking the final 3 days prior to tumor implantation and averaging the values. The percent change is then calculated for each day from baseline for each mouse. The data were then analyzed using a repeated measures two-way ANOVA.

### Food disappearance

Statistical analysis by repeated measures ANOVA.

### Weight

Statistical analysis by repeated measures ANOVA.

### Tumor growth curves

Statistical analysis of tumor growth curves by repeated measures ANOVA.

### Kaplan Meier survival

Statistical analysis of survival by Log-rank (Mantel-Cox) test.

## Acknowledgements

We thank the Histology and Imaging Core (Sanford Research, supported by National Institutes of Health, National Institute of General Medical Sciences, Center of Biomedical Research Excellence 5P20GM103548 and P30GM145398), specifically Claire Evans who provided her services and expertise towards this project. We also thank the following funding sources for their critical contributions to this work: National Institutes of Health, National Institute of Dental and Craniofacial Research grant 5R01DE032712-02 (PDV), Institutional Development Award (IDeA) from the National Institute of General Medical Sciences of the National Institutes of Health grant 5P20GM103548 (PDV). DaCCoTA Scholar Award supported by the National Institute of General Medical Sciences of the National Institutes of Health grant U54 GM128729 (PDV). National Institute of Health, National Cancer Institute grant R37 1R37CA242006-01A1 (MA). Stiefel family Discovery award (MA). Institutional Research Grant (MA). Disruptive Science Moonshot award, MDACC (MA). Canadian Institutes of Health Research grants 162211, 461274, 461275 (ST). National Institutes of Health (R01 CA193522; R21 NS130712) (RD).

# Additional information

## Funding

| Funder | Grant reference number | Author |
|---|---|---|
| National Institute of Dental and Craniofacial Research | 5R01DE032712-02 | Paola D Vermeer |
| National Institute of General Medical Sciences | 5P20GM103548 | Paola D Vermeer |
| National Institute of General Medical Sciences | U54 GM128729 | Paola D Vermeer |
| National Cancer Institute | 1R37CA242006-01A1 | Moran Amit |
| Moonshot Research and Development Program | MDACC | Moran Amit |
| Canadian Institutes of Health Research | 162211 | Sebastien Talbot |
| Canadian Institutes of Health Research | 461274 | Sebastien Talbot |
| Canadian Institutes of Health Research | 461275 | Sebastien Talbot |
| National Institutes of Health | R01 CA193522 | Robert Dantzer |
| National Institutes of Health | R21 NS130712 | Robert Dantzer |
| Stiefel family | Discovery Award | Moran Amit |

The funders had no role in study design, data collection and interpretation, or the decision to submit the work for publication.

## Author contributions

Jeffrey Barr, Conceptualization, Data curation, Formal analysis, Validation, Investigation, Visualization, Methodology, Writing – review and editing; Austin Walz, Data curation, Formal analysis, Validation, Investigation, Visualization, Methodology, Writing – review and editing; Anthony C Restaino, Data curation, Formal analysis, Investigation, Methodology, Writing – review and editing; Moran Amit, William C Spanos, Writing – review and editing; Sarah M Barclay, Investigation, Writing – review and editing; Elisabeth G Vichaya, Data curation, Formal analysis, Writing – review and editing; Robert Dantzer, Sebastien Talbot, Data curation, Formal analysis, Methodology, Writing – review and editing; Paola D Vermeer, Conceptualization, Resources, Data curation, Formal analysis, Supervision, Funding acquisition, Visualization, Methodology, Writing – original draft, Project administration, Writing – review and editing

## Author ORCIDs

Moran Amit ⓘ https://orcid.org/0000-0002-9720-7766
Robert Dantzer ⓘ https://orcid.org/0000-0001-9399-6107
Sebastien Talbot ⓘ https://orcid.org/0000-0001-9932-7174
Paola D Vermeer ⓘ https://orcid.org/0000-0003-2370-8223

## Ethics

Strict accordance with the recommendations in the Guide for the Care and Use of Laboratory Animals of the National Institutes of Health were followed for all animals experiments in this study. Animals were handled according to our approved Institutional Care and Use Committee (IACUC) protocol (#2023-0105). Sanford Research has an Animal Welfare Assurance on file with the Office of Laboratory Animal Welfare (assurance number: A-4568-01) and is accredited by AAALAC, Intl. Sanford Health is also a licensed research facility under the authority of the United States Department of Agriculture (USDA) with USDA certificate number 46-R-011. All surgical procedures (including tumor implantation) were performed on anesthetized animals using ketamine/xylazine.

Reviewer #1 (Public Review): https://doi.org/10.7554/eLife.97916.3.sa1
Reviewer #2 (Public Review): https://doi.org/10.7554/eLife.97916.3.sa2
Reviewer #3 (Public Review): https://doi.org/10.7554/eLife.97916.3.sa3
Author response https://doi.org/10.7554/eLife.97916.3.sa4

## Additional files

### Supplementary files

• Supplementary file 1. List of mouse genes assayed on the ScienCell Gene Query Neuronal Transmission and Membrane Genes plate.

• Supplementary file 2. Significant effects for the statistics presented in *Figure 4*.

• Supplementary file 3. Significant effects for the statistics presented in *Figure 6*.

• MDAR checklist

### Data availability

All data generated or analyzed during this study are included in the manuscript and supporting files.

The following dataset was generated:

| Author(s) | Year | Dataset title | Dataset URL | Database and Identifier |
|---|---|---|---|---|
| Barr J, Walz A, Restaino AC, Amit M, Barclay SM, Vinchaya EG, Spanos WC, Dantzer R, Talbot S, Vermeer PD | 2024 | GeneQuery RNA Profile of Trigeminal Ganglia from Tumor Burden and Tumor Naïve Mice | https://www.ncbi.nlm.nih.gov/geo/query/acc.cgi?acc=GSE275481 | NCBI Gene Expression Omnibus, GSE275481 |

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
