## [Editor Report · eLife assessment]

This **important** research describes the sensory innervation of oral tumors, with potential implications for understanding cancer-induced alterations in motivation and anhedonia in a mouse model. These findings are **solid** and are supported by anatomical and transcriptional changes in the tumor that suggest sensory innervation, neural tracing, and neural activity measurements. While nerve innervation of the tumor and associated increase in brain activity is well-supported, future studies could enhance specificity by employing more targeted genetic and pharmacological tools to manipulate these circuits selectively.

---

## [Referee Report · Reviewer #1 (Public Review)]

Summary:

Using a mouse model of head and neck cancer, Barr et al show that tumor-infiltrating nerves connect to brain regions via the ipsilateral trigeminal ganglion, and they demonstrate the effect this has on behavior. The authors show that there are neurites surrounding the tumors using a WGA assay and show that the brain regions that are involved in this tumor-containing circuit have elevated Fos and FosB expression and increased calcium response. Behaviorally, tumor-bearing mice have decreased nest building and wheel running and increased anhedonia. The behavior, Fos expression, and heightened calcium activity were all decreased in tumor-bearing mice following nociceptor neuron elimination.

Strengths:

This paper establishes that sensory neurons innervate head and neck cancers and that these tumors impact select brain areas. This paper also establishes that behavior is altered following these tumors and that drugs to treat pain restore some but not all of the behavior. The results from the experiments (predominantly gene and protein expression assays, cFos expression, and calcium imaging) support their behavioral findings both with and without drug treatment.

Comments on previously identified weaknesses:

The authors have addressed the majority of my concerns.

---

## [Referee Report · Reviewer #2 (Public Review)]

Summary:

Cancer treatments are not just about the tumor - there is an ever-increasing need for treating pain, fatigue, and anhedonia resulting from the disease as patients are undergoing successful but prolonged bouts with cancer. Using an implantable oral tumor model in the mouse, Barr et al describe neural infiltration of tumors, and posit that these nerve fibers are transmitting pain and other sensory signals to the brain that reduce pleasure and motivation. These findings are in part supported by anatomical and transcriptional changes in the tumor that suggest sensory innervation, neural tracing, and neural activity measurements. Further, the authors conduct behavior assays in tumor-bearing animals and inhibit/ablate pain sensory neurons to suggest involvement of local sensory innervation of tumors in mediating cancer-induced malaise.

Strengths:

• This is an important area of research that may have implications for improving the quality of life of cancer patients.

• The studies use a combination of approaches (tracing and anatomy, transcriptional, neural activity recordings, behavior assays, loss-of-function) to support their claims.

• Tracing experiments suggest that tumor-innervating afferents are connected to brain nuclei involved in oral pain sensing. Consistent with this, the authors observed increased neural activity in those brain areas of tumor-bearing animals. It should be noted that some of these brain nuclei have also been implicated in cancer-induced behavioral alterations in non-head and neck tumor models.

• Experiments are well-controlled and approaches are validated.

• The paper is well-written and the layout was easy to follow.

Weaknesses/Future Directions:

• The main claim is that tumor-infiltrating nerves underlie cancer-induced behavioral alterations. While the studies are supportive of this conclusion, manipulations in the current study are non-specific, ablating all TRPV1 sensory neurons. A direct test would be to selectively inhibit/ablate nerve fibers innervating the tumor or mouth region.

---

## [Referee Report · Reviewer #3 (Public Review)]

Summary:

The authors have tested for and demonstrated a physical (i.e., sensory nerves to brain) connection between tumors and parts of the brain which can provide some clues into why there is an increase in depressive disorders in HNSCC patients. While connections such as this have been suspected, this is a novel demonstration pointing to sensory neurons that is accompanied by a remarkable amount of complementary data.

Strengths:

There is substantial evidence provided for the hypotheses tested. The data are largely quite convincing.

Weaknesses:

The authors mention in their Discussion the need for additional experiments. that address some of the gaps in this analysis.

---

## [Author Response]

The following is the authors’ response to the original reviews.

**Public Reviews:**

**Reviewer #1:**
(1) Study suggests that the effects of their tumor models of mouse behavioral are largely non-specific to the tumor as most behaviors are rescued by analgesic treatment. So, most of the changes were likely due to site-specific pain and not a unique signal from the tumor.

The tumor generates pain at the site it is implanted, and it is likely amplified by the oral activities tumor bearing mice have to engage in. As there is no pain in the absence of the tumor, the pain is, by definition, caused by the tumor, not by the site. Concerning the relationship between pain and behavior, the behavioral assays undertaken in our study (nesting, cookie test, wheel running) were very limited in scope. Two of these assays (nesting, cookie test) require use of the oral cavity. Only nesting and wheel running were assessed in the context of treatment for pain. Nesting behavior was completely restored with carprofen and buprenorphine treatment suggesting that in the absence of pain, mice were able to make perfect nests. Consistent with this, carprofen and buprenorphine treated animals also gained weight indicating that eating (another activity dependent on the oral cavity) was also restored. Wheel running, an activity that does not rely on the oral cavity, was only partially restored with drug treatment. While additional behavioral tests are necessary to confirm this finding, the data suggest that there is pain-independent information relayed to the brain which accounts for this decline in wheel running.

**Reviewer #2:**
(1) The main claim is that tumor-infiltrating nerves underlie cancer-induced behavioral alterations, but the experimental interventions are not specific enough to support this. For example, all TRPV1 neurons, including those innervating the skin and internal organs, are ablated to examine sensory innervation of the tumor. Within the context of cancer, behavioral changes may be due to systemic inflammation, which may alter TRPV1 afferents outside the local proximity of tumor cells. A direct test of the claims of this paper would be to selectively inhibit/ablate nerve fibers innervating the tumor or mouth region.

We agree with the reviewer that a direct test of the hypothesis would require selectively inhibiting the nerve fibers innervating the tumor and assessing the impact on behavior. Studies in the lab are on-going using pharmacological interventions to do this. These studies are beyond the scope of this current manuscript.

(2) Behavioral results from TRPV1 neuron ablation studies are in part confounded by differing tumor sizes in ablated versus control mice. Are the differences in behavior potentially explained by the ablated animals having significantly smaller tumors? The differences in tumor sizes are not negligible. One way to examine this possibility might be to correlate behavioral outcomes with tumor size.

As suggested by the reviewer, we have graphed nesting scores and time-to-interact (cookie test) relative to tumor volume. In both cases, we used simple linear regression to fit the data and analyzed the slopes of the lines. In the case of nesting, there was no significant difference between the slopes. This is now included as Supplemental Figure 4A. In the case of the cookie test, there was a significant difference between the slopes. This is now included as Supplemental Figure 4B. Graphing the data in this way allows one to look at any given tumor volume and infer what the nesting score and the time-to-interact for the two groups of mice. The linear regression model fits the time to interact with the cookie reasonably well, thus from this graph, we can see that at any given tumor volume the time to interact with the cookie was generally shorter in TRPV1cre::DTAfl/wt animals as compared to C57BL/6 mice. Unfortunately, the linear regression does not fit the nesting data very well and thus it is more difficult to make the comparison of tumor volume and nesting score.

The following text has been added to the results section.

Given the impact of nociceptor neuron ablation on tumor growth, we wondered whether differences in tumor volume contributed to the behavioral differences we noted. Thus, the behavior data were graphed as a function of tumor volume (Supplemental Fig 4A, B). A simple linear regression model was used to fit the data. In the case of nesting scores, the linear regression did not fit the data points very well making it difficult to assess nesting scores at a given tumor volume (Supplemental Fig 4A). However, the linear regression model fit the time to interact data better. Here, the graph suggests that tumor volume did not influence behavior as at any given tumor volume the time to interact with the cookie is generally smaller in TRPV1-Cre::Floxed-DTA animals as compared to C57BL/6 animals (Supplemental Fig 4B).

**Reviewer #3:**
(1) The authors mention in their Discussion the need for additional experiments. Could they also include / comment on the potential impact on the anti-tumor immune system in their model?

The following text has been added to the discussion:

Neuro-immune interactions have been studied in the context of a variety of conditions including, but not limited to infection 109, inflammation 110,111, homeostasis in the gut 112-114, as well as neurological diseases115,116. Neuro-immune communications in the context of cancer and behavior have also been studied (e.g., sickness behavior, depression) 117-119 however, these studies did not assess these interactions at the tumor bed. Investigations into neuro-immune interactions occurring within primary malignancies which harbor nerves have shed light on these critical communications. In the context of melanoma, which is innervated by sensory nerves, we identified that release of the neuropeptide calcitonin gene related peptide (CGRP) induces immune suppression. This effect is mediated by CGRP binding to its receptor, RAMP1, which is expressed on CD8+ T cells 49. A study utilizing a different syngeneic model of oral cancer similarly found an immune suppressive role for CGRP 120-122. These studies demonstrate that neuro-immune interactions occur at the tumor bed. Our current findings indicating that tumor-infiltrating nerves connect to a circuit that includes regions within the brain suggest that neuro-immune interactions within the peripheral malignancy may contribute to the behavioral alterations we studied.

(2) The authors mention the importance of inflammation contributing to pain in cancer but do not clearly highlight how this may play a role in their model. Can this be clarified?

The following text has been added to the discussion section of the manuscript.

Moreover, given that carprofen and buprenorphine decrease inflammation 104, their ability to restore normal nesting and cookie test behaviors (which require the use of the oral cavity where the tumor is located) suggests that inflammation at the tumor site contributed to the decline in these behaviors in vehicle-treated animals. Since both drugs were given systemically and each only partially restored wheel running, it suggests that systemic inflammation alone cannot fully account for the decline in wheel running seen in vehicle-treated animals. We posit that the inflammation- and pain-independent component of this behavioral decline is mediated via the transcriptional and functional alterations in the cancer-brain circuit.

(3) The tumor model apparently requires isoflurane injection prior to tumor growth measurements. This is different from most other transplantable types of tumors used in the literature. Was this treatment also given to control (i.e., non-tumor) mice at the same time points? If not, can the authors comment on the impact of isoflurane (if any) in their model?

Mice in all groups (tumor and non-tumor) were treated with isoflurane. This important detail has been added to the methods section.

(4) The authors emphasize in several places that this is a male mouse model. They mention this as a limitation in the Discussion. Was there an original reason why they only tested male mice?

The following text has been added in the discussion section:

Head and neck cancer is predominantly a cancer in males; it occurs in males three times more often than in females 123, this disparity increases in certain parts of the world. While smoking cigarettes and drinking alcohol are risk factors for HPV negative head and neck squamous cell carcinoma, even males that do not smoke and drink are have a higher susceptibility for this cancer than females 124,125. Thus, our studies used only male mice. However, we do recognize that females also get this cancer. In fact, female patients with head and neck cancer, particularly oral cancer, report more pain than their male counterparts 126,127. These findings suggest that differences in tumor innervation exist in males and females.

Therefore, another project in the lab has been to compare disease characteristics (including innervation and behavior) in male and female mice. The findings from this second study are the topic of a separate manuscript.

**Recommendations For The Authors:**

**Reviewing editor:**
(1) Tumors can communicate with the brain via blood-borne agents from the tumor itself or immune cells that are activated by the tumor in addition to neurons that invade the tumor. The xia and malaise that accompanies some tumors can be mediated by direct innervation and/or the humoral factors because both can activate the same parabrachial pathway. This paper makes the case for the direct innervation being important but ignores the possibility of both being involved. The interesting observation that innervation supports tumor growth (perhaps via substance P) is troublesome because the slower appearance of behavioral consequences (Figures 4 & 5) could be attributed to the smaller tumor size. A nice control for humoral effects would be to implant the tumor cells someplace in the body where innervation does not occur (if possible) and then examine behavioral outcomes.

In the course of several projects, we have implanted different tumor cell lines in different locations in mice (oral cavity, hind limb, flank, peritoneal cavity). In each location, tumor innervation occurs. This is not a phenomenon found only in mice as we completed an immunohistological survey of human cancers from different sites and found they are all innervated (PMID 34944001). These data are consistent with tumor and locally-released factors that recruit nerves to the tumor bed (PMID: 30327461)(PMID: 32051587)(PMID: 27989802). Thus, an implantation site that does not result in tumor innervation is currently unknown and likely does not exist.

(2) The authors should address whether there is an inflammatory component in this tumor model.

MOC2-7 tumors have been characterized as non-inflamed and poorly immunogenic 129-131.

This information has been added to the methods section.

(3) The RTX experiment in Figure 5 would be more compelling if the drug was injected directly into the tumor rather than injecting it in the flank, thus ablating all TRPV1-exressing neurons as in the genetic approach.

While we agree with the reviewer that ablating the TRPV1-expressing neurons at the tumor site directly would be ideal, RTX treatment takes approximately one week for ablation to occur but a significant amount of inflammation is associated with this. Therefore, we wait a total of 4 weeks for the inflammation to resolve. By this time, tumors have generally reached sacrifice criteria. Thus, this approach would not enable the question to be answered Moreover, we are not aware of any studies in which RTX has been injected in the oral cavity or face. While RTX is utilized clinically to treat pain, it is typically administered intrathecally, epidurally or intra-ganglionically (PMID: 37894723).

(4) The authors address affective aspects of pain but do not adequately address the sensory aspects, e.g., sensitivity to touch, heat and/or cold. They attribute the decrease in food disappearance (consumption) and nest building to oral pain, but it could be due to anhedonia and anorexia that can accompany tumor progression.

Assaying for touch and heat/cold sensitivity in the oral cavity is a critical aspect of studying head and neck cancer that needs to be addressed. However, in rodents these assays are not trivial given that any touch/heat/cold in the area of the tumor (oral cavity) impacts the sensitive whiskers in that region which directly influence these assays. Thus, we have been refining assays (e.g., OPAD, facial von Frey) to address these important questions. The findings from these studies are beyond the scope of this manuscript.

The reviewer makes a good point about anhedonia and anorexia. The following text has been added to the results section:

Pain-induced anhedonia is mediated by changes in the reward pathway. Specifically, in the context of pain, dopaminergic neurons in the ventral tegmental area (VTA) become less responsive to pain and release less serotonin. This decreased serotonin results in disinhibition of GABA release; the resulting increased GABA promotes an increased inhibitory drive leading to anhedonia 82 and, when extreme, anorexia. Carprofen and buprenorphine treatments completely reversed nesting behavior and significantly improved eating. Inflammation 83 and opioids 84 directly influence reward processing and though our tracing studies did not indicate that the tumor-brain circuit includes the VTA, this brain region may be indirectly impacted by tumor-induced pain in the oral cavity. Thus, an alternative interpretation of the data is that the effects of carprofen and buprenorphine treatments on nesting and food consumption may be due to inhibition of anhedonia (and anorexia) rather than, or in addition to, relieving oral pain.

(5) Comment on why only males were used in this study.

Please see response to public reviews.

**Reviewer #1:**
(1) Please provide a justification for the use of exclusively male mice and expand in the discussion if there is potential for these findings to be directly applicable to female mice as well.

Please see response to public reviews.

The following text has been added to the discussion:

Head and neck cancer is predominantly a cancer in males; it occurs in males three times more often than in females 123, this disparity increases in certain parts of the world. While smoking cigarettes and drinking alcohol are risk factors for HPV negative head and neck squamous cell carcinoma, even males that do not smoke and drink are have a higher susceptibility for this cancer than females 124,125. Thus, our studies used only male mice. However, we do recognize that females also get this cancer. In fact, female patients with head and neck cancer, particularly oral cancer, report more pain than their male counterparts 126,127. These findings suggest that differences in tumor innervation exist in males and females.

(2) When discussing the results shown in Figure 2, please include some mention of Fus, since it was the highest expressed transcript.

The following text has been added to the results section regarding Fus.

The gene demonstrating the highest increase in expression, Fus, was of particular interest; it increases in expression within DRG neurons following nerve injury and contributes to injury-induced pain 51,52. Of note, we purposefully used whole trigeminal ganglia rather than FACS-sorted tracer-positive dissociated neurons to avoid artificially imposing injury and altering the transcript levels of these cells 53,54. Thus, significantly elevated expression of Fus by ipsilateral TGM neurons from tumor-bearing animals suggests the presence of neuronal injury induced by the malignancy. This is consistent with our previous findings 55 and those of others 56 showing that tumor-infiltrating nerves harbor higher expression of nerve-injury transcripts and neuronal sensitization.

(3) In line 197 please clarify the mice used. Were all mice tumor-bearing and some had nociceptors ablated, or was there a control (no tumor) group as well?

Line 197 refers to Figure 4D. In this figure, panels B-D show quantification of cFos and DFosB in the spinal nucleus of the TGM (SpVc), The parabrachial nucleus (PBN) and the Central nucleus of the amygdala (CeA). These data are from C57BL/6 and TRPV1cre::DTAfl/wt animals all of whom had tumor. Supplementary Figure 3C also show quantification of cFos and DFosB but these are from control, non-tumor bearing animals. The fact that controls are non-tumor-bearing has been added to the supplemental figure legend and the text of the results section has been clarified as follows.

While Fos expression was similar between non-tumor bearing mice of the two genotypes (Supplemental Fig. 3C-E), the absence of nociceptor neurons in tumor-bearing animals decreases cFos and DFosB in the PBN, and DFosB in the SpVc (Fig. 4B, C).

(4) Overall it would improve the readability of the figures if the colors for the IHC channels were on the image itself and not exclusively in the figure legend.

The colors for all the staining have been added to each panel.

(5) It is not a problem that complete cartography was not done, but please include a justification for why the brain regions that were focused on were chosen.

In order to ensure that our neural tracing technique captured only nerves present within the tumor bed, we restricted the injection of tracer to only 2 µl. We demonstrated that this small volume did not leak out of the tumor (Figure 1) and thus any tracer labeled neurons we identified were deemed as being connected in a circuit to nerves in the tumor bed. While we acknowledged that this calculated technical approach restricted our ability to tracer label all neurons in the tumor bed (as well as those they share circuitry with), it ensured no tracer leakage and inadvertent labeling of non-tumoral nerves. In non-tumor animals injected with 10 µl of tracer, labeled regions in the brain included the spinal nucleus of the trigeminal, the parabrachial nucleus, the central amygdala, the facial nucleus and the motor nucleus of the trigeminal. The regions that were tracer positive when tumor was injected were limited to the spinal nucleus of the trigeminal, the parabrachial nucleus and the central amygdala. Thus, the regions in the brain that we focused on were the areas that became tracer-positive following injection of tracer into the tumor.

(6) Were the cells that were injected cultured in media with 10% fetal calf serum? If so was any inflammatory response seen? If not please state in the methods section the media that cells for injection were cultured in.

The cells injected into animals were cultured in media containing 10% fetal calf serum. When cells are harvested for tumor injections, they are first washed two times with PBS and then trypsinized to detach the cells from the plate. Cells are collected, washed again with PBS and resuspended with DMEM without serum; this is what is injected into animals. We harvest cells in this way in order to eliminate any serum being injected into mice. This information has been added to the Methods section.

(7) Would any of the differences in drug treatment (Carprofen vs Buprenorphine) be due to the differing routes of administration and metabolism of the drugs?

Since carprofen and buprenorphine each resulted in similar behavioral impacts (nesting and wheel running), their different routes of administration seem to play a minor or no role in the behaviors assessed.

(8) Please include in the methods section the specific approach and software that was used for processing calcium imaging data and calculating a relative change in fluorescence.

The specific approach used for processing calcium imaging data and calculating relative change in fluorescence as well as the software used are all included in the methods section. Please see below:

Ca2+ imaging. TGM neurons from non-tumor and tumor-bearing animals (n=4-6 mice/condition) were imaged on the same day. Neurons were incubated with the calcium indicator, Fluo-4AM, at 37°C for 20 min. After dye loading, the cells were washed, and Live Cell Imaging Solution (Thermo-Fisher) with 20 mM glucose was added. Calcium imaging was conducted at room temperature. Changes in intracellular Ca2+ were measured using a Nikon scanning confocal microscope with a 10x objective. Fluo-4AM was excited at 488 nm using an argon laser with intensity attenuated to 1%. The fluorescence images were acquired in the confocal frame (1024 × 1024 pixels) scan mode. After 1 min of baseline measure, capsaicin (300nM final concentration) was added. Ca2+ images were recorded before, during and after capsaicin application. Image acquisition and analysis were achieved using NIS-Elements imaging software. Fluo-4AM responses were standardized and shown as percent change from the initial frame. Data are presented as the relative change in fluorescence (DF/F0), where F0 is the basal fluorescence and DF=F-F0 with F being the measured intensity recorded during the experiment. Calcium responses were analyzed only for neurons responding to ionomycin (10 µM, positive control) to ensure neuronal health. Treatment with the cell permeable Ca2+ chelator, BAPTA (200 µM), served as a negative control.

(9) Suggestions for Figure 1:- In Figures 1C, D, E, include labels for the days of tumor harvest.- Please make the size of the labels the same for 1K an 1L and align them.- Microscopy image in Figure 1L for SpVc looks like it may be at a different magnification.- If possible, include (either in the figure or the supplement) IHC images staining for Dcx and tau, which would complement the western blot data.

The requested changes to the figures have been made. Unfortunately, we do not have Dcx and tau IHC staining of the day 4, 10 and 20 tumors.

(10) Suggestions for Figure 2:- Include directly onto the graph in Figure 2a the legend for tumor-bearing (red) and non-tumor bearing (blue).- Keep consistent between Figure 2G and 2H/I if the tumor/nontumor will be labeled as T/N or Tumor/Control.

The requested changes to the figures have been made.

(11) Suggestions for Figure 3:- An example trace of calcium signal would complement Figure 3G, H well.

Example tracings of calcium signal are already provided in Supplementary Figure 3A and B.

**Reviewer #2:**
(1) While the use of male mice is acknowledged, there is not a rationale for why female mice were not included in the study.

Please see the response to Reviewer #1 (first question).

(2) Criteria for euthanasia should be described in the Methods. This is especially needed for interpreting the survival curve in Figure 4H.

Criteria for euthanasia in our IACUC approved protocol include:

- maximum tumor volume of 1000mm3

- edema

- extended period of weight loss progressing to emaciation

- impaired mobility or lesions interfering with eating, drinking or ambulation

- rapid weight loss (>20% in 1 week)

- weight loss at or more than 20% of baseline

In addition to tumor size and weight loss, we use the body condition score to evaluate the state of animals and to determine euthanasia. These details have been added to the Methods section.

(3) At what stage in cancer progression were the Fos studies conducted for Figure 4A-D?

The brains used for Fos staining (Fig 4B-D) were harvested at week 5 post-tumor implantation.

(4) For Fos counts, what are the bregma coordinates for the sections that were quantified?

SpVc: -7.56 to -8.24mm

PBN: -4.96 to -5.52mm

CeA: -0.82mm to -1.94mm

(5) Statistics are needed for the claim in Lines 171-173.

The statistical analysis of Fos staining from tumor-bearing and non-tumor bearing brains are included in Figure 3D-F. The statistical analysis of ex vivo Ca+2 imaging of brains from tumor-bearing and non-tumor bearing animals are included in Figure 3 I and J.

(6) How long was the baseline period for weight and food intake measurements? How long were the animals single-housed before taking the baseline measurements?

Baseline weight and food intake measurements were 2 weeks and animals were singly housed before baseline measurements for 2 weeks (a total of 4 weeks).

Minor:(7) The authors might consider rewording the sentence on lines 59-62, given that it is abundantly clear from rodent studies that both the tumor and chemotherapy are associated with adverse behavioral outcomes.

We have reworded the sentence as follows: The association of cancer with impaired mental health is directly mediated by the disease, its treatment or both; these findings suggest that the development of a tumor alters brain functions.

(8) Line 212 needs a space between the two sentences.

This has been fixed.

(9) Font size in Figure 2 is not consistent with the other figures.

This has been fixed.

(10) "DAPI" is the more conventional than "DaPi".

This has been fixed.

Editorial Comments and Suggestions:(1) The Abstract would be better if it were more concise, e.g. ~175 words.

The abstract has been shortened as requested and now reads:

Cancer patients often experience changes in mental health, prompting an exploration into whether nerves infiltrating tumors contribute to these alterations by impacting brain functions. Using a mouse model for head and neck cancer and neuronal tracing we show that tumor-infiltrating nerves connect to distinct brain areas. The activation of this neuronal circuitry altered behaviors (decreased nest-building, increased latency to eat a cookie, and reduced wheel running). Tumor-infiltrating nociceptor neurons exhibited heightened calcium activity and brain regions receiving these neural projections showed elevated cFos and delta FosB as well as increased calcium responses compared to non-tumor-bearing counterparts. The genetic elimination of nociceptor neurons decreased brain Fos expression and mitigated the behavioral alterations induced by the presence of the tumor. While analgesic treatment restored nesting and cookie test behaviors, it did not fully restore voluntary wheel running indicating that pain is not the exclusive driver of such behavioral shifts. Unraveling the interaction between the tumor, infiltrating nerves, and the brain is pivotal to developing targeted interventions to alleviate the mental health burdens associated with cancer.

(2) Lines 28, 104, 258, 486, 521, and many other places, "utilized" should be "used" because the former refers to an application for which it is not intended, e.g. a hammer was utilized as a doorstop.

The requested changes have been made.

(3) Lines 32 and 73, it is not clear whether the basal activity is heightened or whether excitability is increased. "manifest" might be better than "harbor" on line 73.

We have changed the wording in the abstract to be clearer. Moreover, our finding that TGM neurons from tumor-bearing animals have increased expression of the s1-Receptor and phosphorylated TRPV1 (Fig 2G-I) indicate that these neurons have increased excitability.

(4) Line 34 and elsewhere, it would be better to refer to Fos because the is no need to distinguish cellular, cFos, from viral, vFos, in this context.

The requested changes have been made.

(5) Line 38, It would be better to refer to what was actually measured rather than "oral movements".

The requested changes have been made. The sentence now reads: “While analgesic treatment restored nesting and cookie test behaviors, it did not fully restore voluntary wheel running.”

(6) Line 84, CXCR3-null mouse on a C57BL/6 background.

The requested change has been made.

(7) Lines 86,129 wild-type, male mice.

The requested change has been made.

(8) Lines114-115, the brackets are not necessary.

The requested change has been made.

(9) Lines 118, 384, 409, 527, 589, 971, 974 always leave a space between numbers and units. Use Greek u for micro.

The requested change has been made.

(10) Lines 123-124, it is not clear that there is meaningful labeling within the CeA.

We have replaced this image with a more representative one of the CeA from a tumor-bearing animal with clear tracer labeling.

(11) Lines 125, 138, and 246 transcription was not measured, only transcript levels were measured.

The requested changes have been made.

(12) Line 133, I think >4 fold is meant.

Thank you for catching that. I have fixed it to >4 fold.

(13) Line 165, single-time-point assessment (add hyphens).

The requested change has been made.

(14) Line 181 and elsewhere including figure, the superscripts refer to alleles of the genes; hence approved gene names should be used in italics (as in Methods), TRPV1-Cre:: Floxed-DTA (without italics) would be acceptable.

The requested changes have been made.

(15) Line 182, nociceptor-neuron-ablated mice (add hyphens).

The requested changes have been made.

(16) Line 197, It is not clear that the "speed" of food disappearance was measured or that it is due to oral pain vs loss of appetite.

The reviewer makes a good point. We have changed the sentence to read:

To evaluate the effects of this disruption on cancer-induced behavioral changes, we assessed the animals’ general well-being through nesting behavior 32 and anhedonia using the cookie test 76,77, as well as body weight and food disappearance as surrogates for oral pain and/or loss of appetite.

(17) Line 199, The reduced tumor growth after ablation could account for most of the changes in the other parameters that were measured.

We have graphed the nesting scores and time-to-interact with the cookie as a function of tumor volume. These data are now included as Supplemental Figure 4 and suggest that at the same tumor volume, nesting scores and times-to-interact with the cookie are different between the groups.

(18) Line 204 TPVP1 spelling. Is the TGN smaller after ablation of half of the neurons?

The requested change has been made.

(19) Line 235, "now" is not necessary.

The requested change has been made.

(20) Line 238-239 and elsewhere, a few references for to why the TGN-SpVc-PBN-CeA circuit is relevant would be helpful.

The following references have been added regarding the relevance of this circuit to behavior:

Molecular Brain 14: 94 (2021) (PMID 34167570)

Neuropharmacology 198: 108757 (2021) (PMID 34461068)

Frontiers in Cellular Neuroscience 16: 997360 (2022) (PMID 36385947)

Neuropsychopharmacology 49(3): 508-520 (2024) (PMID 37542159)

(21) Lines 371, 434 and Figures, gm should be g or grams in scientific usage. Include JAX lab stock numbers for these mouse lines.

The requested changes have been made.

(22) Line 432, removing food for one hour is not a fast.

The sentence has been reworded as follows: One hour prior to testing, mouse food is removed and the animals are acclimated to the brightly lit testing room.

(23) Line 476, 5-um sections (add hyphen).

The hyphen has been added.

(24) Lines 988, and 1023, DAPI are usually shown this way.

The requested change has been made.

(25) Figure 1K, add Bregma levels to figures.

SpVc: -8.12 mm

PBN: -5.34 mm

CeA: -1.34 mm

(26) Figure 3 line 1033, "area under the curve" What curve was examined?

The curve examined was the change in fluorescence over time. This curve has been added as Supplemental Figure 3C.

(27) Figure 3B, the circled area is the lateral PBN. At first glance, I thought scp was meant as the label for the circled area.

Scp is noted in the figure legend as a landmark.